# Integrating cell morphology with gene expression and chemical structure to aid mitochondrial toxicity detection

Srijit Seal [1], Jordi Carreras-Puigvert [2], Maria-Anna Trapotsi[1], Hongbin Yang[1], Ola Spjuth [2✉] & Andreas Bender [1✉]

Mitochondrial toxicity is an important safety endpoint in drug discovery. Models based solely on chemical structure for predicting mitochondrial toxicity are currently limited in accuracy and applicability domain to the chemical space of the training compounds. In this work, we aimed to utilize both -omics and chemical data to push beyond the state-of-the-art. We combined Cell Painting and Gene Expression data with chemical structural information from Morgan fingerprints for 382 chemical perturbants tested in the Tox21 mitochondrial membrane depolarization assay. We observed that mitochondrial toxicants differ from non-toxic compounds in morphological space and identified compound clusters having similar mechanisms of mitochondrial toxicity, thereby indicating that morphological space provides biological insights related to mechanisms of action of this endpoint. We further showed that models combining Cell Painting, Gene Expression features and Morgan fingerprints improved model performance on an external test set of 244 compounds by 60% (in terms of F1 score) and improved extrapolation to new chemical space. The performance of our combined models was comparable with dedicated in vitro assays for mitochondrial toxicity. Our results suggest that combining chemical descriptors with biological readouts enhances the detection of mitochondrial toxicants, with practical implications in drug discovery.

[1] Yusuf Hamied Department of Chemistry, University of Cambridge, Lensfield Rd, Cambridge CB2 1EW, UK. [2] Department of Pharmaceutical Biosciences and Science for Life Laboratory, Uppsala University, Box 591, SE-75124 Uppsala, Sweden. ✉email: ola.spjuth@farmbio.uu.se; ab454@cam.ac.uk

Drug-induced mitochondrial toxicity[1] is being increasingly recognized as a contributor to late-stage withdrawals[2] causing cardiotoxicity[3], drug-induced liver toxicity[4] and diseases related to ageing such as Parkinson[5]. Mitochondrial toxicity can be directly or indirectly caused by combinations of multiple mechanisms (some of the mechanism are shown in Fig. 1 although this in no way is an exhaustive list) which makes predicting mitochondrial toxicity challenging[6]. There are multiple factors affecting mitochondrial toxicity. A common direct cause of mitochondrial dysfunction is uncoupling of the electron transport chain from ATP synthesis or accumulation of calcium in mitochondria causing an increase in Reactive Oxygen Species (ROS) leading to oxidative stress and damaging mitochondrial DNA (mtDNA)[7]. Indirect effects of drugs on cells such as inhibition of fatty acid β-oxidation, uncoupling of oxidative phosphorylation, the opening of the membrane permeability transition pore, and disruption of mtDNA synthesis and translation have also been shown to cause mitochondrial toxicity[7]. Retrograde signalling pathways often triggered by these mechanisms result in cross-talk between mitochondria and nucleus leading to changes in nuclear gene expression and may activate unfolded protein response[8]. Besides ER stress, oxidative stress, proteotoxic stress, and apoptosis, there are also other types of signalling and outcomes in aspects of mitochondrial organelle biology such as mitophagy and mitochondrial-derived vesicles[9] that may contribute to mitochondrial toxicity although the outcome may be cell death when there is exessive cytotoxicity. Hence, we can see that mitochondrial toxicity can be difficult to predict with just chemical structure and there is a need to include more biological data which may be more predictive of this endpoint.

The risk of mitochondrial toxicity in drug discovery can be indicated either via experimental methods (such as the Glu/Gal assay[10]) or using predictive methods trained on data from in vitro assays. Dedicated assays often use HepG2 cells to detect mitochondrial toxicants[11] and mostly fluorescent dyes (Mito-MPS[12], DiOC6[13], rhodamine-123[14], MitoTracker Orange[15], TMRM[14],

TMRE[14], JC-1[16]). These assays capture proxy endpoints, for example, membrane depolarisation, which are quite heterogeneous and not in absolute concordance to the term "mitochondrial toxicity". Another example, in the Glu/Gal assay, the ratio of $IC_{50}$ values in different cultures is not always easy to translate to in vivo effects. Further, each fluorescent dye has its limitation[17], for example, JC1 is sensitive to membrane depolarization but disadvantaged by its poor water solubility and low signal-to-background window. Imaging assays, such as the Apredica HepG2 mitochondrial membrane potential and mitochondrial mass assays[18] measure the average cell intensities for mitochondria from high content imaging, where the average intensity of mitochondria is used to define mitochondrial membrane potential, and the total intensity is used to define mitochondrial mass[18]. This makes defining and detecting mitochondrial toxicity a challenging task in itself.

Previous approaches to computationally predict mitochondrial toxicity has to a large extent been based on predicting mitochondrial membrane depolarisation using chemical structure (Fig. S1, Supplementary Data 1) and machine learning methods including Support Vector Machines[19], Random Forest models[20,21], and naïve Bayes classifier[22]. Using molecular descriptors or structural fingerprints, the best models showed a balanced accuracy between 0.74 to 0.86 as reported by Zhao et al.[20] However, Zhao et al. showed that extrapolation to new structural space is difficult and accuracy inside the model's applicability domain was higher when compared with out-of-domain compounds[20].

Since mitochondrial toxicity can be characterised by a multitude of mechanisms[3], it has been challenging to assemble sufficient data that can sustain computational methods able to extrapolate to new chemical space. Together with the fact that in vitro assays for mitochondrial toxicity are demanding and with varying degrees of reliability, there is a clear need for advancements in the field[23]. In recent years, hypothesis-free data on cell lines have become available on a much larger scale, both publicly and in company repositories. In this work, we explore how data from Cell Morphology in addition to Gene Expression can

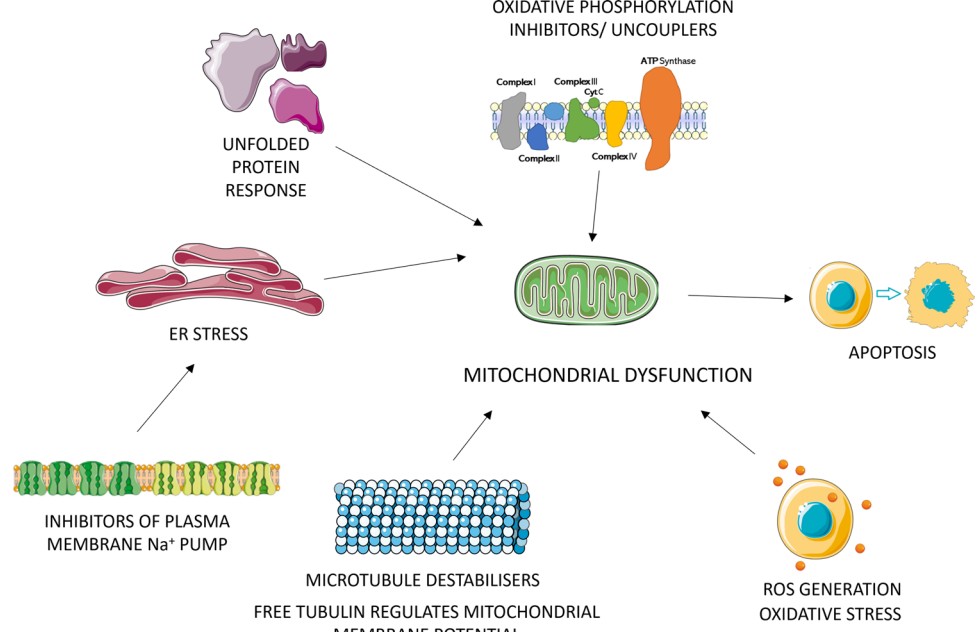

**Fig. 1 Some of the major mechanisms of action of mitochondrial toxicants.** Toxicants act on multiple pathways to exhibit mitochondrial toxicity, including but not limited to, unfolded protein response, inhibition of mitochondrial respiratory chain or uncoupling of oxidative phosphorylation, oxidative stress from responses including generation of reactive oxygen species (ROS), microtubule disruption and ER stress from various responses including inhibition of Na$^+$ pumps etc. The Figure was partly generated using Servier Medical Art, provided by Servier, licensed under a Creative Commons Attribution 3.0 unported license.

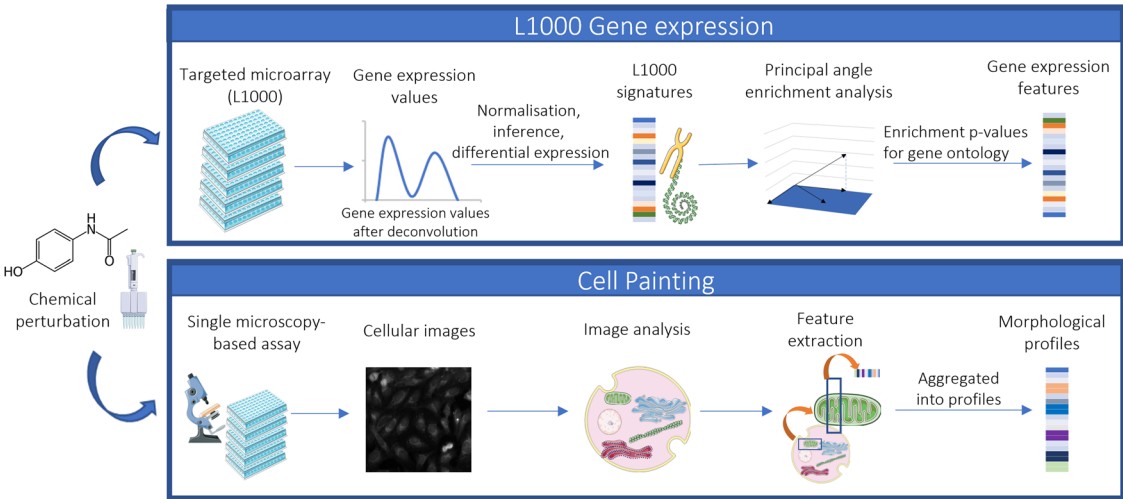

**Fig. 2 Overview of the workflow employed in this study.** L1000 technology for Gene Expression and Cell Painting Technology for cell morphology statistics. The LINCS L1000 gene expression technology profiles changes in 978 landmark genes before and after chemical perturbations on different human cell lines. Raw unprocessed flow cytometry data from Luminex is converted to quantile-normalized gene expression profiles for all replicates of each compound. We use Gene Expression data from Wang et al.[45], who for each compound, computed the strongest gene expression signature using the Characteristic Direction (CD) method and computed enrichment *p* values for each CD signature in the space of all genes against gene set libraries using Principal Angle Enrichment Analysis (PAEA). The Cell Painting assay, on the other hand, captures cellular morphological changes in the form of numerical statistics which are converted from microscopic image data of cells treated with chemical perturbations. The Figure was partly generated using Servier Medical Art, provided by Servier, licensed under a Creative Commons Attribution 3.0 unported license.

improve the detection of mitochondrial toxicity. To the best of our knowledge, this is the first study that presents predictive models for mitochondrial toxicity in vitro assays based on integrated data derived from two types of hypothesis-free data and chemical structure.

The LINCS L1000 gene expression technology developed by Broad Institute (described in Fig. 2) captures changes in 978 landmark genes, and large scale data before and after treating different human cell lines with FDA-approved drugs and small molecules is now available on a sufficiently broad base to be useful for modelling[24]. Gene Expression features have been used in predicting in vitro cell viability[25], drug protein targets[26], and organ level toxicity such as hepatotoxicity[27], nephrotoxicity and cardiotoxicity[28]. The Gene Ontology initiative aims to unify gene and gene product attributes in a classification effort that will provide functional interpretation of gene expression data which, in our case, helps better generalise pathways of mitochondrial toxicity[29]. The Cell Painting assay (described in Fig. 2) is a relatively recent technology developed by the Broad Institute and is used to capture cellular morphological changes in image data from genetic or chemical perturbations[30,31]. Microscopic images are processed to obtain over 1700 measures of cellular and organelle changes such as morphology, texture and intensity. Cell Painting features have been previously used in predicting in vitro toxicity such as cytotoxicity[32], bioactivity endpoints[33], annonymised assay activity[34,35] and mechanism of action[36], cell health phenotypes[37], drug-protein targets[38], antiviral drug discovery[39] as well as organ level toxicity such as drug-induced liver toxicity[40]. Further, it has been also shown that such cell morphology space provides a feature-specific subspace that is complementary information to biological information contained in the gene expression[41], which has also been shown in predicting the mechanism of action of compounds[42]. However, the predictivity of high-dimensional biological features for safety- or efficacy-related endpoint needs to be established in each case, which for in vitro-to-in vivo extrapolation (for example from mitochondrial toxicity to liver injury caused by the former) is not a trivial exercise[43,44].

With the availability of high throughput hypothesis-free data from cell profiling technologies, we are presented with new opportunities to improve the detection of mitochondrial toxicity. In this work, we use Cell Painting and Gene Expression features to extrapolate the applicability domain of structure-based models to the new chemical space. While Gene Expression data is easier to directly interpret[45], in this work we put particular emphasis on exploring and interpreting the biological significance and applicability of Cell Painting features that contain information about mitochondrial toxicity.

## Results

Data for in vitro mitochondrial toxicity was collected from the Tox21 assay[46] for mitochondrial membrane potential disruption summary assay (AID 720637)[47]. Image-based morphological features from the Cell Painting assay were collected from Bray et al.[31] Gene Expression features were extracted from the LINCS L1000 dataset as pre-processed by the Ma'ayan Lab[45]. A combined dataset was assembled to be used for model development (henceforth referred to as "training data") that contained 404 distinct compounds (62 mitotoxic, 320 nontoxic and 22 inconclusive) that contained both Cell Painting features and Gene Expression features. An external test set was assembled that comprised a total of 244 distinct compounds (47 mitotoxic and 197 nontoxic) from Hemmerich et al.[21] who compiled various assays relevant to the toxicity of mitochondrial function, binding and inhibition and an additional 8 compounds from Mitotox Database[48] (which was released towards the end of us conducting this study) which were not covered by the former. No compound in this external test set of 244 compounds overlapped with the training data. Both datasets covered drugs over a wide range of ATC code distribution at the top level 327 drugs (training data) and 111 drugs (external test set) as shown in Fig. S2.

**Mitochondrial toxicants are similar in morphological space.** We analysed if mitochondrial toxicants were more similar to each other in morphological space than toxicants to non-toxicants,

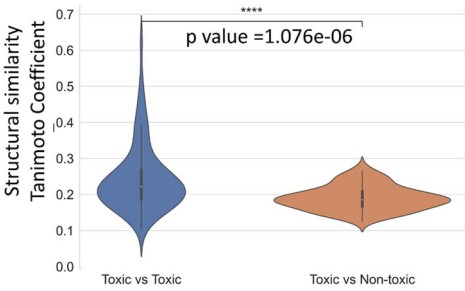

**Fig. 3 Mitotoxic compounds are considerably different from non-toxic compounds in morphological space.** Intra- and inter-class pairwise similarity for 486 compounds (85 mitotoxic) Median of five highest Tanimoto similarity coefficients of Morgan fingerprints and (B) Absolute value of the median of fifteen most positively and fifteen most negatively Pearson correlation effect sizes of selected 110 Cell Painting features for mitochondrial toxic and non-toxic compounds. Mitotoxic compounds considerably vary from non-toxic compounds in morphological space (median Pearson correlation of 0.140 vs 0.038, $t$ test independent $p$ value = 3.301e–20) while also varying in structural space (median Tanimoto Similarity of Morgan fingerprints 0.208 vs 0.183, $t$ test independent samples $p$ value = 6.329e–03).

which could be a prerequisite for the use of this readout space for the detection of mitochondrial toxicity. This was done by comparing the median values of the 5 highest Tanimoto similarity coefficients and the absolute value of median of 15 most positively and 15 most negatively Pearson correlation statistic values for Cell Painting features. As shown in Fig. 3, we found that mitotoxic compounds are considerably different from non-toxic compounds in morphological space (median Pearson correlation of 0.08 vs 0.01, $t$ test independent $p$ value = 3.3e–20). However, they remain distinguishable in structural space (median Tanimoto Similarity of Morgan fingerprints 0.22 vs 0.19, $t$ test independent samples $p$ value = 6.3e–03). We conclude that morphological space can discriminate between mitochondrial toxicants and non-toxicants and that this readout space is more able to discriminate between both classes of compounds than chemical fingerprints on the dataset analysed here.

**Cell Painting features cluster mitochondrial toxicants to identify different mechanisms of mitochondrial toxicity.** We firstly analysed the morphological readout space for the ability to differentiate between mechanisms of action (MOA) for mitochondrial toxicity. We performed feature selection on the initial 1,729 features (see "Methods Section") which selected 110 Cell Painting features and visualised the morphological space using Principal Component Analysis (PCA). As shown in Fig. 4, compound clusters emerged, which were related to mitochondrial toxicity (for further details see Supplementary Data 2). In particular, Cluster I (Fig. 4) comprises several microtubule destabilisers such as fenbendazole, parbendazole, and mebendazole, that belong to the benzimidazole class[49–51] together with structurally dissimilar compounds, namely rotenone and paclitaxel, both of which are known mitochondrial toxicants as well as microtubule destabilizers[52,53]. Figure S3 shows that Cell Painting phenotypes for six microtubule disruptor drugs (Cluster I: albendazole, colchicine, mebendazole, paclitaxel, parbendazole and podophyllotoxin) reveal alterations at the nuclear level, depicted by nuclear fragmentation as well as multinucleated cells, vacuolation of the endoplasmic reticulum, redistribution of the mitochondria and cytoskeleton destabilisation. We found ouabain and digoxin in Cluster II (Fig. 4) have similar mechanisms for mitochondrial injury as inhibitors of the plasma membrane $Na^+$ pump, which can lead to impaired mitochondrial $Ca^{2+}$ retention, increased ROS production and reduced mitochondrial membrane potential[54,55]. Cluster III (Fig. 4) consists of statins, namely lovastatin and simvastatin, which are known to inhibit the synthesis of mevalonate, a precursor of ubiquinone that is vital to the mitochondrial respiratory chain and causes oxidative stress[56].

Compounds in Cluster IV (Fig. 4), namely mevastatin (a statin), raloxifene (a selective estrogen receptor modulator) and prazosin (an alpha-blocker), form again a cluster that is rather diverse with respect to chemical structures, and primary pharmacology/indication areas. However, those compounds are all known to induce apoptotic signalling cascades which trigger the release of cytochrome c into the cytosol[57–59]. This causes depolarization in the mitochondrial membrane leading to mitochondrial injury. Although compounds in the individual clusters were often structurally dissimilar to each other, we did not find any other compound in the training dataset (542 compounds) with a very similar chemical structure (greater than 0.85 Tanimoto similarity) to the compounds in the individual mitochondrial toxicity clusters. This shows morphology space could cluster dissimilar structures with similar modes of action together and did not miss similar compounds with similar modes of action. Overall, our findings show that cell morphology readouts from the Cell Painting assay can cluster several modes of action of mitochondrial toxicants, such as the disruption of microtubules, increased ROS production and oxidative stress.

**Cell Painting features are correlated to Gene Expression features.** We used 62 known mitotoxic compounds to calculate Pearson's correlation between the selected 110 Cell Painting features, and 10 Gene Expression features related to unfolded protein response, endoplasmic reticulum stress, T cell apoptotic process and side of the membrane which represent biological processes from prior knowledge is known to be related to mechanisms mitochondrial toxicity[60,61]. We found specific Cell Painting features were correlated with these Gene Expression features as shown in Fig. 5 (further details on biological significance in Supplementary Data 3). We found that Gene Expression features corresponding to unfolded protein response and endoplasmic reticulum stress were most positively correlated to "Cytoplasm_AreaShape_FormFactors". Form factors indicate how perfectly circular an object is which corresponds to the rounding up of cells due to apoptosis and could be indicative of cell death caused by ER stress or the unfolded protein response which induces cell death, like many other stress responses[62]. Gene Expression features related to unfolded protein response were negatively correlated to "Cells_Texture_DifferenceVariance_RNA_10_0" which calculates the image variation in a normalized co-occurrence matrix and could correspond to various secondary processes following ER stress (including a reduction in transcription, but also reduced translation, caspase activation, apoptosis, etc.[63]). The Gene Expression feature "side of membrane", which is a parent to the cytoplasmic side of mitochondrial outer membrane was found most positively

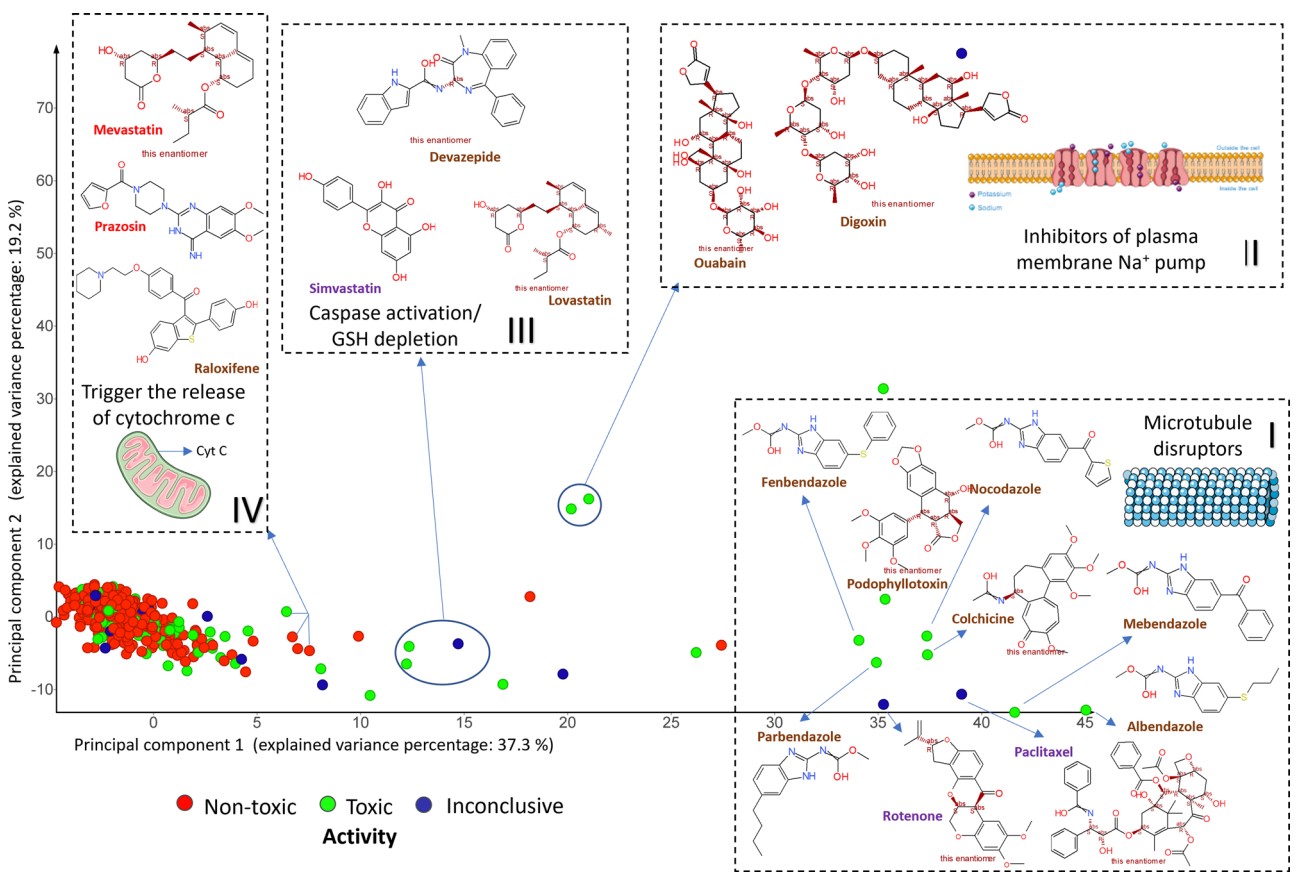

**Fig. 4 Compound having similar mechanisms of action cluster in morphological space.** Principal Component Analysis of 542 compounds in 110-dimensional Cell Painting feature space. Certain compounds clustered further away from the distribution of majority of compounds having similar mechanisms of actions such as of microtubule destabilizers, or compounds inducing apoptotic signalling cascades, compounds causing oxidative stress due to GSH depletion or those that are inhibitors of plasma membrane Na$^+$ pump (all of which reduce mitochondrial membrane potential). Cluster I (microtubule destabilisers): rotenone, albendazole, parbendazole, mebendazole, nocodazole, fenbendazole, colchine, paclitaxel and podophyllotoxin; Cluster II (inhibitors of plasma membrane Na$^+$ pump): ouabain and digoxin; Cluster III (caspase activation and GSH depletion): devazepide, lovastatin, simvastatin; and Cluster II (trigger the release of cytochrome c into the cytosol): mevastatin, prazosin, and raloxifene. The Figure was partly generated using Servier Medical Art, provided by Servier, licensed under a Creative Commons Attribution 3.0 unported license.

correlated to "Nuclei_Granularity_1_RNA" and most negatively correlated to "Cytoplasm_Correlation_Costes_DNA_Mito". An increase or decrease in granularity of cytoplasmic RNA, in the proximity of the nucleus, might indicate the formation of RNA inclusion bodies or RNA processing while the correlation between DNA and mitochondria object could correlate to DNA fragmentation and heterogeneity in mitochondrial content. Hence, we conclude that Cell Painting features contain information of biological significance related to pathways of mitochondrial membrane depolarisation.

**Cell painting and Gene Expression enables training of accurate and interpretable models for detecting mitochondrial toxicity**. As the utility of chemical structure in detecting mitochondrial toxicity was previously explored by Hemmerich et al.[21], our work focussed on comparing individual Cell Painting features and Gene Expression features with respect to their ability to detect mitochondrial toxicants. We used positive predictive values (PPV) and F1 scores from single decision tree classifiers trained on individual features (see "Methods") to detect a signal for mitochondrial toxicants and provide a biological interpretation of these feature spaces. We found that Cell Painting features related to granularity, intensity, location, and radial distribution of mitochondrial objects over the three compartments (cells, cytoplasm and nuclei) had high predictivity for mitochondrial toxicity

(median PPV grouped by compartment, channel, and feature group greater than 0.70; Fig. S4). We next more closely considered the feature value distribution for individual features with high PPV for mitotoxicity (Fig. S5). For example, in "Cells_Intensity_MaxIntensityEdge_Mito" (PPV = 0.83) compounds toxic to mitochondria evenly affect the edge of the mitochondria object. Since this measurement is at the edge of the segmented object, it indicates a loss of membrane integrity. Another feature, "Cells Intensity MADIntensity Mito" (PPV = 0.8) is a measurement of statistical dispersion which measures the standard deviation and median absolute deviation (MAD) of pixel intensity values while being robust to outliers. For MitoTracker Deep Red used in Cell Painting assay, this might indicate a variation of intensities among fragments of the mitochondrial membrane, resulting from loss of membrane integrity. "Cells Granularity 1 RNA" (PPV = 0.56) reveals information present in pixel 1 size in the RNA channel where certain mitotoxic compounds also have low feature values. An increase or decrease in granularity of cytoplasmic RNA might indicate the formation of RNA inclusion bodies or RNA processing. Further attempted biological interpretations for some features (knowing that this is not a trivial process) are shown in Fig. 6 and Supplementary Data 4.

Gene Expression features with high PPV could be classified as either causing mitochondrial membrane depolarisation or as an effect of mitochondrial toxicity (as shown in Fig. 7 and

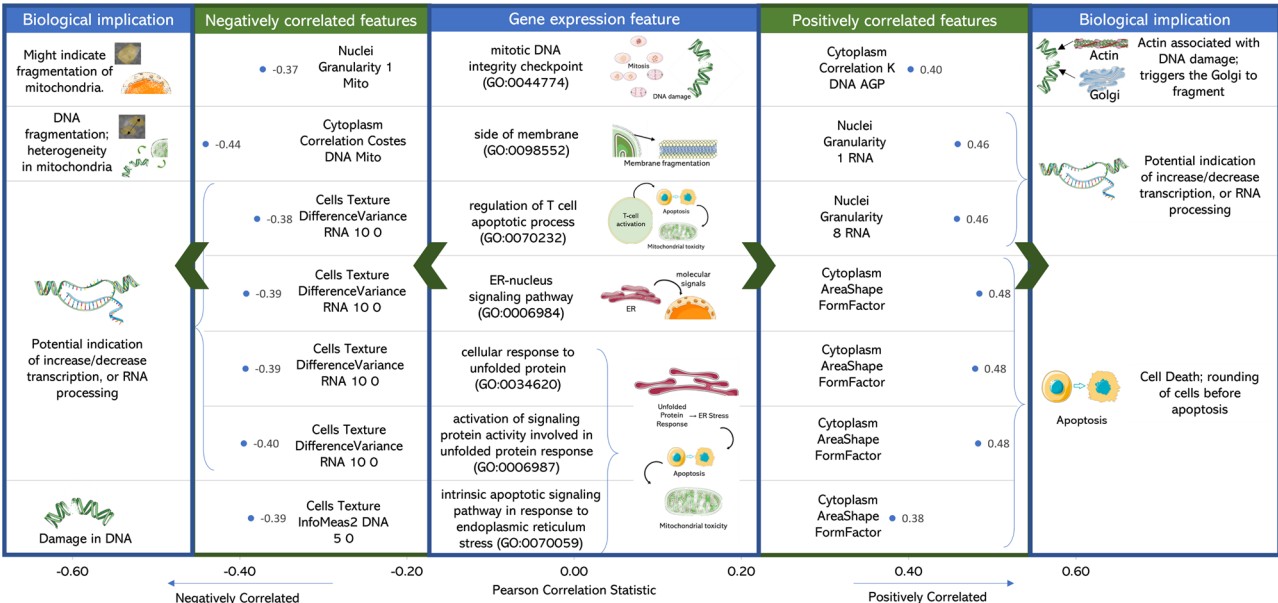

**Fig. 5 Biological implication in mitochondrial toxicity of the cell painting features correlated to gene expression.** Computational significance and biological implication in mitochondrial toxicity of the Cell Painting features that are most positively or negatively correlated to Gene Expression descriptors particularly unfolded protein response and endoplasmic reticulum stress (RNA variance and cell area shape), T cell apoptotic processes (mitochondrial granularity and DNA fragmentation) and side of the membrane (RNA granularity and heterogeneity in mitochondria). Further details in Supplementary Data 3. The Figure was partly generated using Servier Medical Art, provided by Servier, licensed under a Creative Commons Attribution 3.0 unported license.

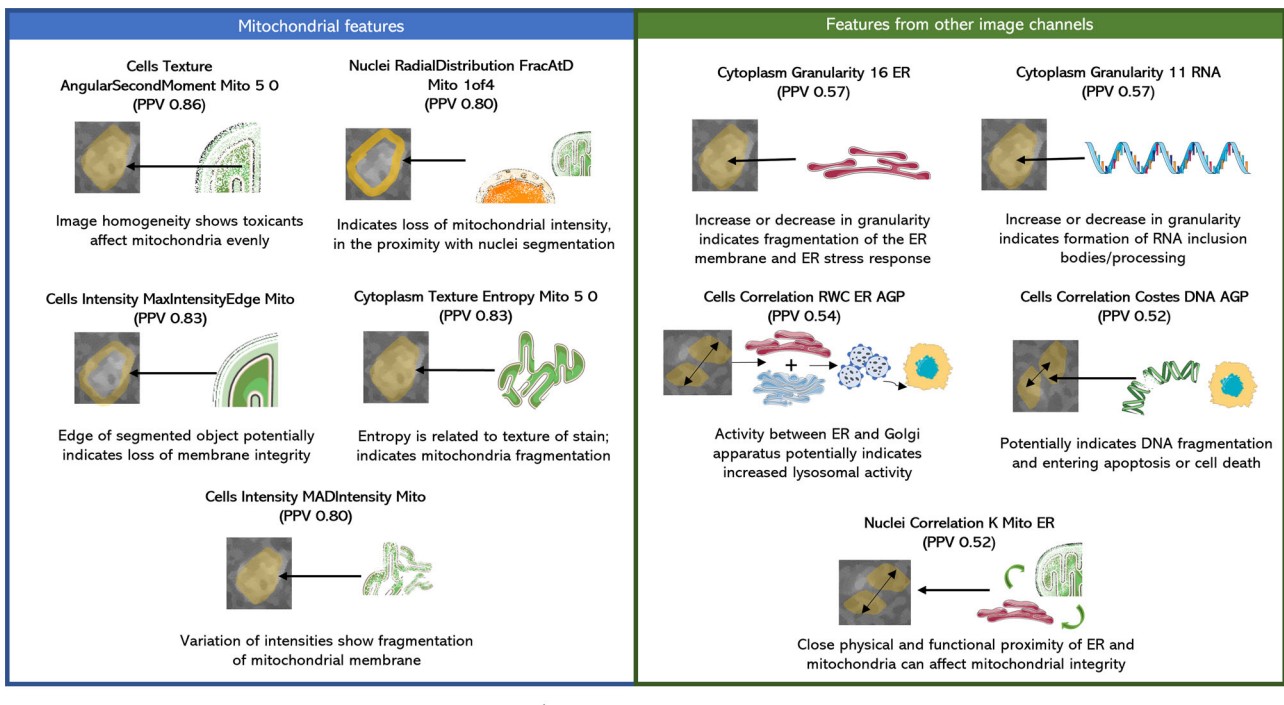

**Fig. 6 Biological implication of Cell Painting features in relation to mitochondrial toxicity.** Biological implication of mitochondrial toxicity translated from the computational image statistics of Cell Painting features. Features were mainly related to edge intensity of cells (possibly related to integrity of cell wall), radial distribution and intensity in mitochondria (related to mitochondrial death) and granularity features (related to cell death and amount of information contained in cellular images). Further details in Supplementary Data 4. AGP Actin Golgi Plasma membrane, DNA Deoxyribonucleic acid, ER endoplasmic reticulum, Mito Mitochondria, RNA Ribonucleic acid. The Figure was partly generated using Servier Medical Art, provided by Servier, licensed under a Creative Commons Attribution 3.0 unported license.

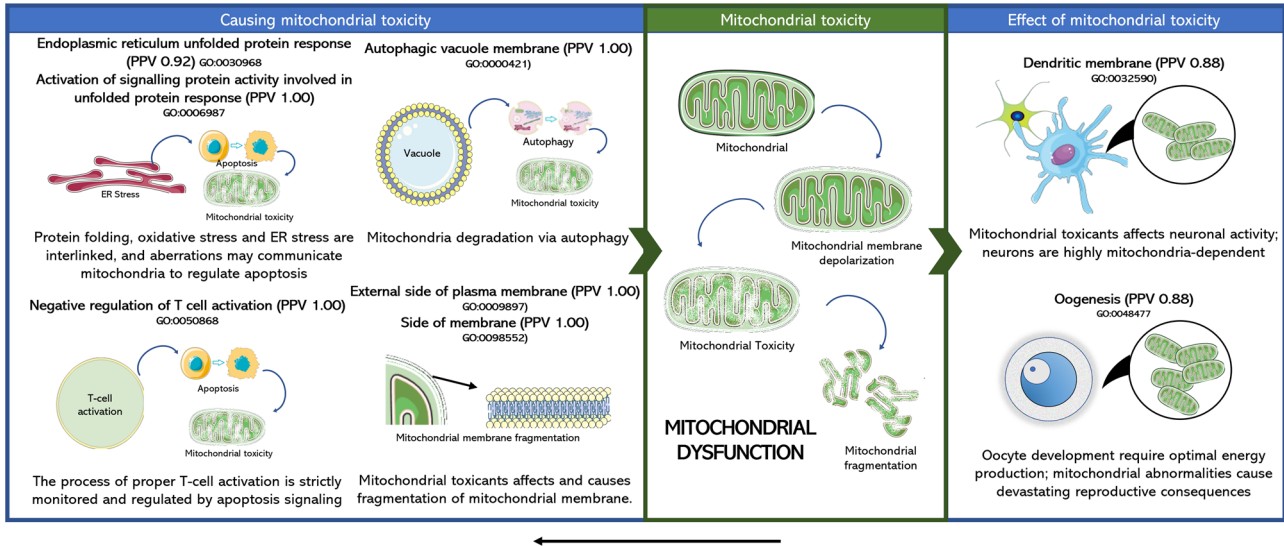

**Fig. 7 Biological implication of gene expression features in relation to mitochondrial toxicity.** Biological implication of mitochondrial toxicity translated from the Gene Expression features. Features causing mitochondrial toxicity mainly related to unfolded protein response (possibly related to ER stress) and plasma membrane (related to membrane depolarisation). Some effects of mitochondrial toxicity were also captured by Gene Expression features such as oogenesis and dendritic plasma membrane; both processes are heavily mitochondria dependent. Further details in Supplementary Data 5. The Figure was partly generated using Servier Medical Art, provided by Servier, licensed under a Creative Commons Attribution 3.0 unported license.

Supplementary Data 5). Features such as endoplasmic reticulum unfolded protein response (PPV 0.92) and activation of signalling protein activity involved in unfolded protein response (PPV 1.00) have previously been attributed to ER related effects such as protein folding, oxidative stress and ER stress[62]. Such effects are linked to each other and toxins affecting the same can depolarise the mitochondrial membrane affect the movement of mitochondria on microtubules and eventually regulate apoptosis[64]. Features such as the external side of plasma membrane (PPV 1.00), side of membrane (PPV 1.00), autophagic vacuole membrane (PPV 1.00), negative regulation of T cell activation (PPV 0.86) are related to processes of cell proliferation, cell cycle arrest as well as apoptosis that causes oxidative stress and cell death which can cause mitochondria to depolarise[65]. The GO Cellular Component dendritic plasma membrane (PPV 0.88) and the Biological Process oocyte development (PPV 0.88) are greatly affected by mitochondrial dysfunction as neurons are mitochondria-dependent cells[66] while oocyte development requires optimal energy production and is highly dependent on mitochondrial function for the same[67]. Hence we conclude that a number of Cell Painting and Gene Expression features showing a high predictivity to mitochondrial membrane depolarization are also interpretable in the mechanistic roles either causing mitochondrial toxicity or being a consequence of the same.

**Fusion models accurately detect mitochondrial toxicity and expand the applicability domain.** We finally established predictive models for mitochondrial toxicity based on 3 models using Cell Painting features, Gene Expression features and Morgan fingerprints and another 2 combinations thereof in early- and late-stage fusion. Early-stage fusion appended all three features into a single vector while late-stage fusion averaged the probabilities of the three individual models. We used a Random Forest model with repeated nested cross validation on a training data of 382 compounds (out of which 62 have mitotoxic annotations) and validated using an external dataset of 244 compounds (47 mitotoxic) where test compounds, although run across various assay conditions were structurally diverse and generally dissimilar to the training data

(details shown in Fig. S6). Figure 8 shows median performance from nested-cross validations and external validations (for further results see Supplementary Data 6 and for an overview of results from each fold of nested-cross validation see Fig. S7).

Fusion models combining Cell Painting features, Gene Expression features, and Morgan fingerprints exhibited higher F1 scores on the external dataset (early-stage fusion: 0.47, late-stage fusion: 0.42) in detecting mitotoxicity than models using only Morgan fingerprints (0.25). The drop in the F1 score of models using only Morgan fingerprints from 0.42 in repeated nested cross-validation to 0.25 in the external test (Fig. 8) shows that Morgan fingerprints lack extrapolation power to novel chemical space. Although the training dataset was different and larger in previous work by Hemmerich et al.[21] focusing on purely chemical structure data (1412 compounds vs 382 compounds here), the results are hence not directly comparable; our early-stage fusion model based had slightly higher F1 scores (0.47vs. 0.41) which implied improved ability to detect mitochondrial toxicants in the external test set. As shown in Fig. 9a, the success of fusion models is underlying by the fact that in morphological space, mitotoxic compounds in the external test set were more morphologically similar to mitotoxic compounds in the training set while no such correlation was present among the images of non-toxic compounds. Finally, late-stage fusion was more sensitive to the toxic class in the external test set compared to early-stage models (0.79 vs 0.64) while the balanced accuracy remained the same (0.69). Given the importance of detecting mitotoxicants in practice, the higher sensitivity and F1 score of a model is likely advantageous in practical situations even at identical balanced accuracy.

We next analysed in more detail the predictions of mitochondrial toxicants in the external test set with our models as shown in Fig. 9b (with further details shown in Supplementary Data 7). The model using Morgan fingerprints could correctly classify only 9 out of the 47 mitochondrial toxicants in the external test set; these compounds were at a low structural distance to mitotoxic compounds in the training set (Figs. S6 and S8). The model using only Cell Painting features could extrapolate well into structurally diverse compounds and correctly predict 34 out of 47 mitotoxic compounds in the external test set but failed when the distance to

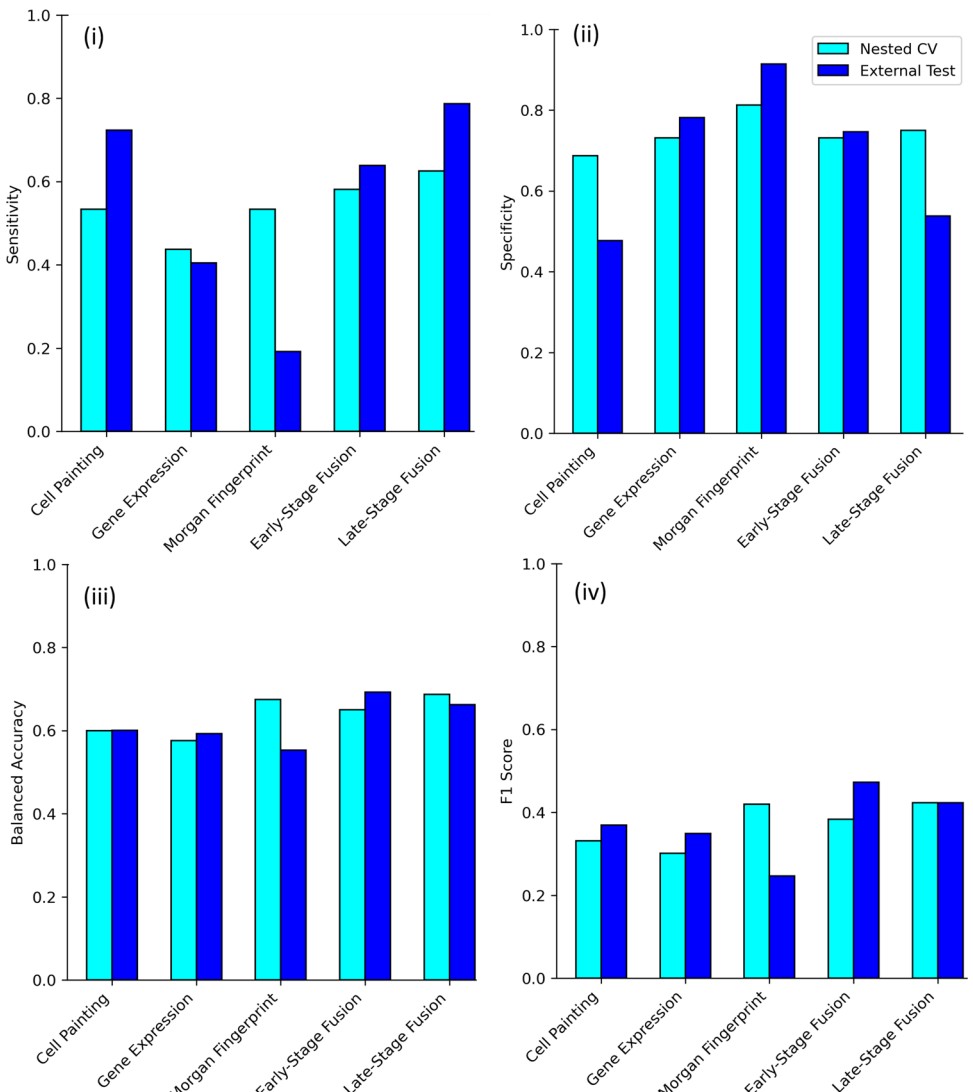

**Fig. 8 Performance of models used in this study from nested-cross validations and external validation.** Evaluation metrics, namely from (i) sensitivity, (ii) specificity, (iii) balanced accuracy and (iv) F1 score for five models from **a** Nested CV (median of repeated nested cross validations) from the training data ($n = 382$ compounds) and **b** external test set ($n = 244$ compounds). Early-stage fusion and Late-stage fusion models combining all three feature sets of Cell Painting, Gene Expression and Morgan have higher F1 score for compounds exhibiting mitochondrial toxicity and extrapolate well into new structural space in external test set compared to models using Morgan fingerprints where F1 Score performance falls by 60% (0.25–0.40 in absolute terms).

morphological space was high for example, with compounds 71145-03-4 (Fig. 9b), while the same compound was correctly predicted by the model using only Morgan fingerprints (which was explicable due to lower structural distance to training data). The late-stage fusion model correctly predicted 37 out of 47 mitotoxic compounds, combining information from both spaces, out of which 5 mitochondrial toxicants were neither correctly predicted by the model using only Cell Painting features, nor by the model using only Morgan fingerprints. Among them were betulinic acid, ketoconazole and diflunisal (which inhibits oxidative phosphorylation[68,69] and fluoxetine (which inhibits oxygen consumption and lowers mitochondrial ATP[70] shown in Fig. 9b)). These examples demonstrate the synergistic effect of the late-stage fusion model, using information from both the cell morphological as well as the chemical fingerprint space.

**Late-stage fusion models accurately detect mitochondrial toxicity of Tox21 compounds labelled as inconclusive.** Next, we compared predictions using the 5 models above for the 22 compounds from the data where inconclusive results were obtained in

the Tox21 due to excessive cytotoxicity either in the mitochondrial depolarization assay or in the cell viability assay[71]. Literature analysis revealed (further details in Supplementary Data 8) that 4 of the 22 compounds (loratadine, progesterone, ticlopidine and tyrphostin A25) previously have been shown to not cause mitochondrial damage (in fact, progesterone[72] and tyrphostin A25[73] reduce oxidative stress and repair oxidative damage). Another 10 compounds showed some mitochondrial toxicity, such as ketoconazole (inhibitor of oxidative phosphorylation[68]), diflunisal (uncoupler of oxidative phosphorylation[74]), daidzein and fipronil (increase ROS causing mitochondrial depolarization[75,76]). The mitochondrial toxicity for the remaining 8 compounds could not be elucidated further from the literature. Mitochondrial toxicity, like any other compound effect, is concentration-dependent, and the literature evidence compiled as well as the Cell Painting assays whose data was used in this work might hence use different concentrations. Also, the cell line/biological system considered in the literature evidence, the Tox21 assays and the Cell Painting assay can also be very different. With respect to concentration, we explored to what extent the data used here would be predictive for

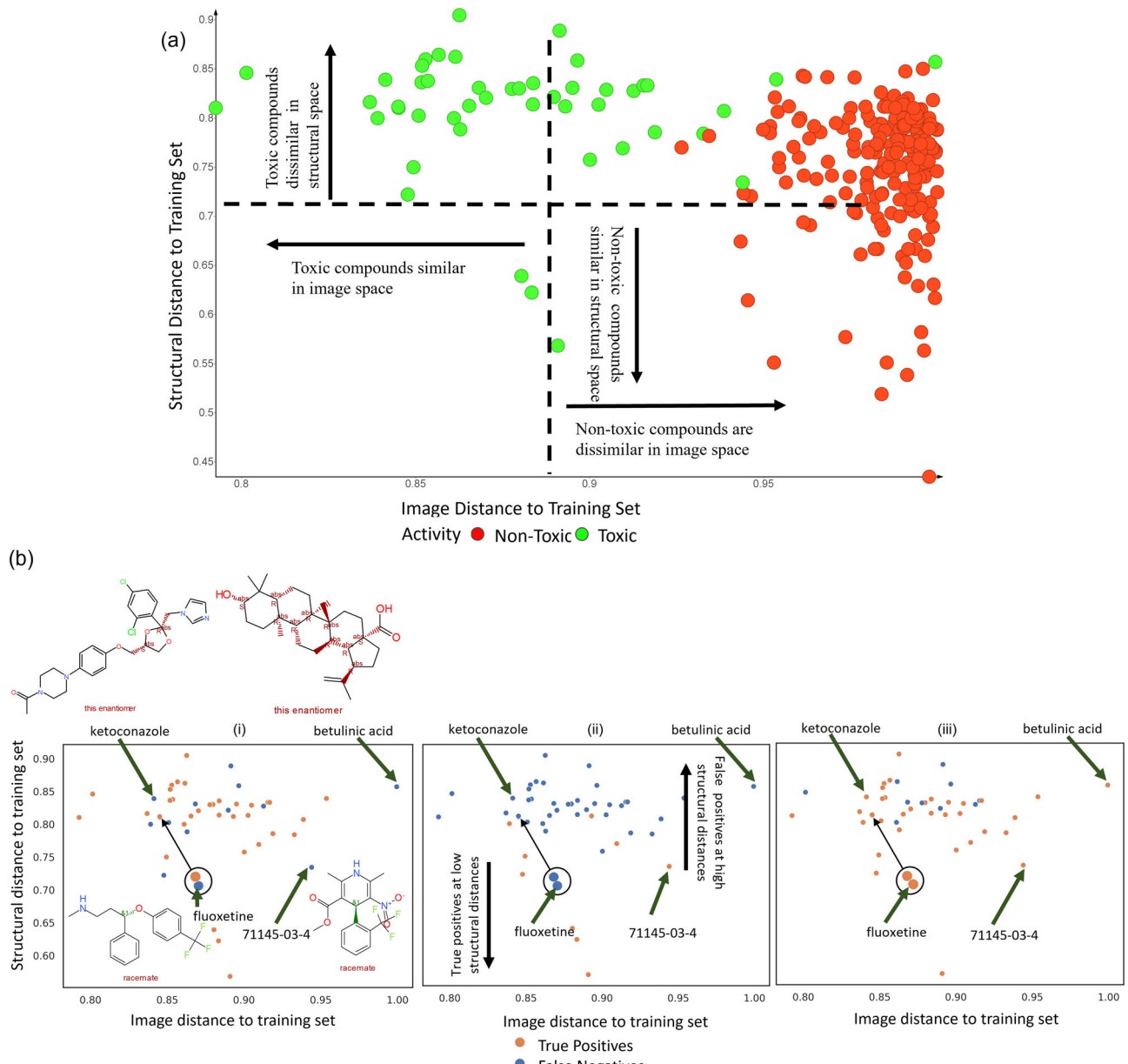

**Fig. 9 Predictions of mitochondrial toxicants in the external test set in relation to structural and morphological distance to training set. a** Most mitotoxic compounds are similar in image space for training ($n = 382$ compounds) and external test set ($n = 244$ compounds), however, non-toxic compounds in the external test set were dissimilar to non-toxic compounds in training set in the image space. Further, toxic compounds are often dissimilar in structural space, indicating the need for fusion models. **b** Structural and morphological distance for mitotoxic compounds in external test set ($n = 244$ compounds) to the training set ($n = 382$ compounds) for models using (i) Cell Painting features, (ii) Morgan fingerprints and the (iii) Late-stage fusion models. Morgan fingerprints failed to correctly classify mitotoxic compounds (eg. betulinic acid) at high structural distances while models using Cell Painting features could extrapolate well into structurally diverse compounds. The late-stage fusion models correctly classified mitotoxic compounds (e.g. 71145-03-4 or methyl 2,6-dimethyl-5-nitro-4-[2-(trifluoromethyl)phenyl]-1,4-dihydropyridine-3-carboxylate, ketoconazole and fluoxetine) in both diverse morphological and structural space where individual models failed demonstrating the synergistic effect of the features spaces.

the mitotoxicity endpoint considered, on a relative scale for the different input parameters used in our models; while for the cell line used it has been shown before that Cell Painting yields similar biological phenotypes for different cell lines without cytochemistry protocols requiring specific cell-type-specific optimization[77]. The latter indicates the predictions from the Cell Painting assay may be applicable in detecting toxicity mitochondrial toxicity in another cell line or biological system.

For the 14 compounds for which mitotoxicity annotations were found, Morgan fingerprints correctly predicted toxicity of only 1 out of 10 toxic compounds and correctly predicted the non-toxic

nature of 3 out the 4 non-toxic compounds (Supplementary Data 9). Thus, Morgan fingerprints showed only very low sensitivity on this dataset. The best performing model, late-stage fusion (averaging predictions from all three models using Cell Painting, Gene Expression features and Morgan fingerprints) however correctly predicted toxicity for all (10 out of 10) mitochondrial toxicants, and correctly predicted the non-toxic nature of 3 out the 4 non-toxic compounds (that is, the increased sensitivity does not come at a cost of a large false positive rate, given that the latter stayed constant between both models). Overall, the late-stage fusion model could hence extrapolate to the

morphological space of these inconclusive compounds and detect mitochondrial toxicity even when Tox21 assays reported inconclusive outcomes due to cytotoxicity.

**Sensitivity of fusion models is on par with dedicated in vitro mitochondrial toxicity assays**. Finally, we compared the performance of our models to detect mitochondrial toxicity with dedicated in vitro assays. Hallinger et al. compared various high throughput screening assays and a respirometric screening assay to detect known mitochondrial toxicants (as shown in Fig. S1 and Supplementary Data 1)[78]. When comparing 60 reference chemicals to existing Tox21 assays, they found RSA to be most predictive (balanced accuracy 0.90), while the Tox21 mitochondrial membrane potential assay was also highly predictive (balanced accuracy 0.87). However, respirometric screens have lower throughput than Tox21 assays and are not suitable for screening a large number of compounds. Among other assays they compared were high content imaging assays, where the Apredica HepG2 mitochondrial membrane potential and mitochondrial mass assays[18] were found to be comparatively less predictive (balanced accuracy 0.78 and 0.65, respectively). Although the 244 compounds in the external test set in our study are not the same as these 60 reference chemicals, from a numerical performance comparison we found that our fusion model achieved sensitivity at par with imaging assays in the external test set (0.79 in our study vs 0.37 in Apredica MitoMass vs 0.8 in RSA) in detecting mitotoxic compounds with comparable balanced accuracies (0.69 in our study vs 0.65 in Apredica MitoMass). The added advantage of using Cell Painting is that it is a comparatively inexpensive single screen that can also be used simultaneously for multiple endpoints for which it is found to be predictive. Hence, we can conclude that the late-stage fusion model based on Cell Painting, Gene Expression and chemical structural data compares well with respect to its predictive power for mitotoxicity to many dedicated assays for this purpose (although precise numerical values cannot be compared due to the different data sets used).

**Limitations and future work**. One limitation of the here presented study is the size of the dataset (overlap between the Cell Painting and Gene Expression features with mitochondrial toxicity assay annotations), making it cover a somewhat limited area of chemical space. To mitigate the risk of overfitting in an ensemble method such as Random Forests, first the Cell Painting and Gene Expression features were subjected to feature selection. We used hyperparameter optimisation and nested cross validation to evaluate the training data and selected the optimal number of trees as shown in previous work[79]. Although an individual decision tree may be more prone to over-fitting, in the case of an ensemble method such as Random Forests, we also avoid over-fitting by bootstrapping samples (randomly choosing selected features at each split for trees)[80]. To further examine this, we used an external test set where we found the performance does drop in the external test set when using only chemical structure but not when using fusion models combining structure with cell morphology and gene expression. Our evaluation and discussions on extrapolation to new chemical space (as shown in Fig. 9) are based on this external test set and hence are still valid.

Another limitation is the discrepancy between the cell lines, where Cell Painting was carried out on USO2 cells, and LINC L1000 used a variety of cell lines (MCF7, A549, HepG2, HT29, etc.); we use the Cell Painting to predict toxicity in the Tox21 assay which used HepG2 as testing cell lines thus giving rise to a toxicity discrepancy in assays using different tissues and perturbants at different concentrations. In our case, we leverage the fact that cell morphological data provides versatile biological data that is generally able to extrapolate to different cell lines as shown in previous studies. For example, Cox et. al. used cell morphological data from 15 reporter cell lines to predict the mechanism of action (MOA) but did not see any individual reporter cell line outperform others (with the notable exception of GR agonists)[81]. They also observed that the genetic background of the reporter cell line did not affect the overall AUC-ROC values calculated for the different MOAs. Although gene expression data can be cell line specific, previous work by Lapins et al showed that the prediction of MOA was similarly effective with an average AUC of 0.83 across 3 different cell types[36]. Therefore as outlined in the *Methods* Section *Gene Expression features* subsection, for gene expression features in this study we used the strongest signatures irrespective of the cell line. In this manner, although the cell line of feature spaces differs from the toxicity assay (and more so in the case of organ-level toxicity), we could leverage the biological information in the cell morphology and gene expression data to predict mitochondrial toxicity which is an in vitro toxicity endpoint.

Future studies would benefit from larger datasets, such as the announced future data depositions from the JUMP consortium[82], and also more and better annotated compounds that show mitochondrial toxicity under different assays and dosages such as from the Mitotox database[48]. It may also be possible to apply different types of machine learning or deep learning models, such as deep neural networks, gradient boosting, or a variational autoencoder which has been previously shown to reveal an interpretable latent space[83] to improve the model's predictions and generally improve the interpretability of models.

## Discussion

Mitochondrial toxicity is a leading cause of late-stage drug withdrawals[2] and numerous drugs such as amiodarone, doxorubicin, statins (e.g cerivastatin, simvastatin) and valproic acid have been shown to induce adverse effects via mitochondrial dysfunction. Mitochondrial toxicity can be caused by multiple mechanisms and prediction using only chemical structural fingerprints has been shown to be difficult, with respect to extrapolation to novel chemical space, where low model sensitivity has been regularly observed.

To the best of our knowledge, in this work, we present the first study combining hypothesis-free high throughput Cell Painting and Gene Expression features with structural fingerprints to predict mitochondrial toxicity. In this work, we confirmed that Cell Painting readouts can discriminate mitotoxic and non-mitotoxic compounds and are able to cluster mitotoxic compounds with a shared mode of action (including compounds with inconclusive assay outcomes in Tox21 due to excessive cytotoxicity) in morphological space. This indicates that Cell Painting features are able to detect similarities with respect to both modes of action and mitochondrial toxicity, also in situations of large differences in chemical space. Further, we showed that Cell Painting features correlate to Gene Expression features, which are related to mechanisms of mitochondrial toxicity. We trained late-stage fusion models, which are averaging the results from the Cell Painting, Gene Expression and Morgan fingerprint models. We show that the late-stage fusion model has higher accuracy when predicting mitochondrial toxicity (F1 score 0.42) when extrapolating to the new chemical space of an external set (wherein compounds which were structurally dissimilar to the training set as shown in Fig. S6) compared to the model using only Morgan fingerprints (F1 score of 0.25). As shown in Fig. 9, the utilization of cell morphology and gene expression data improved the detection of mitotoxic compounds (as shown in late-stage fusion model with sensitivity 0.79 vs

0.19 when using only structural data). Hence, we can conclude that the detection of mitochondrial toxicity is improved when using all three feature spaces (structure, morphology, and gene expression) together. Also compared to dedicated mitochondrial high content imaging assays our late-stage fusion model showed favourable sensitivity. The Cell Painting feature space is less expensive than the L1000 assay or DRUG-seq[84] and thus enables larger high-throughput experiments[42]. This cell morphology modality is thus being increasingly explored both in the public domain such as by the JUMP consortium[82] and as well as by pharmaceutical companies such as by Jannsen[85]. Given that Cell Painting readouts can be used for multiple purposes; this supports their use also for the prediction of a mitochondrial toxicity endpoint.

Using hypothesis-free data, such as Cell Painting and Gene Expression data, in machine learning models can hence be used to detect toxicity (here mitochondrial toxicity), as well as to help understand modes of toxicity, also in situations where this is not possible based on chemical structure alone. From a predictive modelling perspective, by combining high predictivity of fingerprints in areas of structural space close to the training set with better generalizability of Cell Painting features at greater distances to the training set, such models can contribute to extending the applicability domain of the overall model.

## Methods
In this work, we followed the workflow as displayed in Fig. S9 for dataset curation, feature selection and model architecture.

**Mitochondrial toxicity dataset preparation**. Data for in vitro mitochondrial toxicity, used as the endpoint in this study, was collected from the Tox21 assay[46] for mitochondrial membrane potential disruption (MMP) summary assay (AID 720637)[47]. This summary assay combines results from a mitochondrial membrane potential (MMP) assay[86] (AID 720635) and a cell viability counter screen[87] (AID 720634) into a binary assay hit call.

For the Tox21 MMP assay, a water-soluble mitochondrial membrane potential sensor was used to evaluate chemically induced mitochondrial toxicity. In healthy cells, this dye accumulates in the mitochondria with red fluorescence. However, should the potential collapse, the dye is no longer able to accumulate in the mitochondria and remains in monomers giving a green fluorescence from the cytoplasm. The cytotoxicity was tested in the same assay well as the mitochondrial potential using a counter cell viability screen. The viability of the cells in the culture was determined by measuring the amount of ATP present. Thus, the summary assay not only considers triplicate runs of the ratio (red/green) readout in the MMP assay but also each fluorescence channel separately, as well as the cytotoxicity results[71]. The compounds causing excessive cytotoxicity were labelled "inconclusive" which helps differentiate compounds that decreased MMP from those inducing high cytotoxicity. One cannot be certain if mitochondrial dysfunction may have caused the excessive cell death in these "inconclusive compounds". Hence as a precautionary measure to not end with predicting cytotoxicity in this study, but only mitochondrial toxicity, we removed inconclusive compounds from the training dataset. Hence, in our models, inconclusive compounds were removed and for the remaining compounds, mitochondrial toxicity labels were assigned as per assay hit calls from the Tox21 summary assay.

Multiple endpoints, such as mitochondrial membrane depolarisation, can be related to mitochondrial toxicity as can be increased ROS or alteration of energy homeostatic, especially if the membrane potential is depolarised. Hemmerich et al.[21] compiled various mitochondrial membrane potential as well as additional assays relevant to mitochondrial toxicity (mitochondria, mitochondria potential and mitochondria complex) from various sources, including CHEMBL, PubChem and 246 drugs tested by Zhang et al.[19] into a single dataset related to mitochondrial function, binding and inhibition consisting of 824 mitochondrial toxicants and 4937 non-toxic compounds. To evaluate our models, we further used an external test set using compounds (that do not appear in the training data) from this dataset. We further included compounds from Mitotox Database[48] compiling mitochondrial toxicity under different assays and dosages. We searched Mitotox for compounds associated to decreased transmembrane potential[88] to obtain 652 unique mitotoxic compounds.

**Cell painting features**. Image-based morphological features were extracted from the Cell Painting assay experiments in Bray et al.[31] These experiments contained perturbations from 30000 chemicals (around 10,000 small molecules, 2200 drugs and 18000 novel compounds from diversity-oriented synthesis) using DMSO neutral control, USO2 cells in 384-well plates in 5 channels staining eight cellular organelles: nucleus, endoplasmic reticulum, F-actin cytoskeleton, Golgi apparatus, plasma membrane, mitochondria, cytoplasmic RNA and the nucleoli. We obtained consensus morphological features for each compound using the following procedure similar to Lapins et al.[36] For each plate, the average feature value from the DMSO plates was subtracted from the perturbation's average feature value. Next, we calculated the median feature value for each compound and dose combination. For replicates, we used the median feature values for doses that were within one standard deviation of the mean dose. The concentration was also included as a feature. Features known to be noisy and generally unreliable as recommended by Way et al. were removed[89]. Further, the changes in morphology may be particularly obvious due to excessive cytotoxicity there we must avoid perturbations which drastically reduced the cell count compared to the neutral control of DMSO. We removed such compounds (step 1, Fig. S9a) in Cell Painting images by removing compounds with a threshold of 1.5 times standard deviation below the mean of "Cells Number Object Number" (the distribution is shown in Fig. S9b) which is below −15.09. Thus, consensus morphological profiles consisting of 1729 numerical features were obtained.

**Gene expression features**. The Gene Ontology initiative aims to unify gene and gene product attributes in a classification effort that will provide biological and functional interpretation of gene expression data[29]. They also ensure that genes are consistently annotated across different available datasets. The Gene Expression features used in this work have been derived from transcriptomic data from LINCS L1000. LINCS L1000 gene expression technology profiles changes in 978 landmark genes on perturbations of compounds for a variety of human cell lines[24]. In this work, Gene Ontology transformed Gene Expression features were extracted from the http://maayanlab.net/SEP-L1000/#download which contained 4438 annotated Gene Expression features corresponding to 19803 distinct compounds[45]. The authors used quantile-normalized gene expression profiles from the LINCS L1000 dataset for all replicates of each compound. For each compound, the strongest signatures were used irrespective of the cell line, concentration, or time point which minimizes the number of features required. Gene expression signatures for each compound perturbation were computed using the Characteristic Direction (CD) method[90] on 978 measured hallmark genes. Further, they computed enrichment $p$ values for each CD signature in the space of all genes against gene set libraries including biological processes, cellular components, and molecular functions, as well as other gene set libraries accessible from the Enrichr tool[91] using an extension of the CD technique called Principal Angle Enrichment Analysis (PAEA)[92]. We used these annotations for each Gene Expression-perturbation combination for further analysis.

**Dataset curation and collation**. We calculated the intersection between mitochondrial toxicity and Cell Painting and Gene Expression datasets described above using standard InChI calculated using RDKit[93]. For conflicting replicates, we considered a compound toxic, if it was detected to be mitotoxic at least once (since in such situations evidence for mitochondrial toxicity, at least under some conditions, exists). We obtained 830 distinct compounds (161 mitotoxic, 61 inconclusive and remaining non-toxic) from the mitochondrial toxicity dataset overlapped with Gene Expression features and a total of 513 distinct compounds (82 mitotoxic and 27 inconclusive) in overlap with Cell Painting features. Similarly, we found 404 distinct compounds (62 mitotoxic and 22 inconclusive) in overlap with both Cell Painting features and Gene Expression features. For the external test set, after adding required annotations of Cell Painting features and Gene Expression features, removing compounds with low cell count, and ensuring no compounds from this were used in feature selection or training our models, a total of 236 distinct compounds (39 mitotoxic and remaining nontoxic) remained in the external test set. From the Mitotox database we obtained 652 unique compounds, out of which both Cell Painting and Gene Expression data were available for only 71 compounds. 63 out of these 71 compounds were already used in this study (54 were used in the training data and 9 in the external test set) We added the remaining 8 compounds to our external test dataset thus totalling 244 distinct compounds (47 mitotoxic and remaining nontoxic). To evaluate our models, we used this an external test set where compounds in this external test set do not appear in the training data.

**Structural fingerprints**. For modelling purposes, we used Morgan fingerprints which contain structural information about compounds and have been successfully used before for toxicity prediction[94]. The MolVS standardizer, an open-source tool based on RDKit[93], was used to standardize (including tautomer standardization) and canonicalize SMILES of the parent molecules[95]. This involved sanitization, normalisation, greatest fragment chooser, charge neutralisation, tautomer enumeration, and canonicalization as implemented in the MolVS tool and described in the MolVS standardizer. We calculated Morgan fingerprints of radius 2 and 2048 bits from standardized SMILES using RDKit[93].

**Feature selection**. For each of the Cell Painting features and Gene Expression features, standardized values for compounds in the training set were separately subjected to three statistical tests, namely, the two-sample Kolmogorov Smirnov test (KS test)[96], Mann–Whitney U test[97] (MWU test) and Point-Biserial correlation[98] (PBS correlation). While the Random Forest algorithm employed for

modelling (see *Model generation and evaluation* below) is in principle able to select features, this is not always successful which made us compare different explicit feature selection methods in parallel with inputting all features into the models subsequently. This method in our experience led to less overfitting while still being interpretable and able to extrapolate to the external test set compared to other methods (Principal Component Analysis, Maximum Relevance — Minimum Redundancy or using all features, see Supplementary Data 10 for modelling results when comparing different feature selection methods).

After removing the inconclusive compounds (step 2, Fig. S9a), the feature selection (step 3, Fig. S9a) was performed on Cell Painting features and Gene Expression annotations for the remaining compounds. For cross-validation of models, the overlap of the top 40 negatively and 40 positively correlated features from the MWU test and PBS correlation and top 40 correlated features from the KS test were selected for further modelling. For evaluating the external test where more data was available for training, we selected the top 25 correlated features from each test (both positively and negatively for MWU and PBS) and obtained 110 Cell Painting features and 102 Gene Expression features.

**Comparing class separation and visualization of compounds in morphological space**. For a comparison of intra-class (Toxic vs Toxic) and inter-class (Toxic vs Nontoxic) in morphological space, we used 486 compounds (85 mitotoxic) for which Cell Painting annotations were available. We randomly resampled the majority class (non-toxic compounds) to match the number of samples of the minority class to ensure our comparisons are equivalent. Then we visualised mean Tanimoto similarity, median positive image correlation (considering only positive Pearson correlations) and absolute median image correlation (considering the absolute value of median both positive and negative Pearson correlations) for various values of k in k-nearest neighbours in four quartiles of the distribution for intra- and inter-class pairwise distributions (Fig. S10). We found better separation between intra- and inter-class pairwise when using the absolute value of the median values from the most 15 positively and 15 negatively pairwise Pearson correlations of Cell Painting features. For visualizing the same in structural space, we used the median of 5 highest pairwise Tanimoto similarity of Morgan Fingerprints[99,100]. The methodology was followed when comparing tests to train set distances as defined in subsection *"Extrapolation to New Structural/Morphological Space"*.

For visualization of compounds in morphological space, we analysed the 110 selected Cell Painting features on 513 distinct compounds (85 mitotoxic and 27 inconclusive). We normalized 110 selected Cell Painting features and performed Principal Component Analysis using DataWarrior[101] which compared to other nonlinear methods is more interpretable.

**Correlation between cell painting and gene expression features and their positive predictive values**. To determine the correlation between selected Cell Painting features and Gene Expression features for compounds exhibiting mitochondrial toxicity, we used Pearson correlation using the pandas Python package[102]. Comparing the negative logarithmic *p* value and the effect size, we determined which Cell Painting features were correlated to specific Gene Expression features related to unfolded protein response, endoplasmic reticulum stress, and T cell apoptotic process, side of membrane etc.

Random Forests are not able to detect feature importance when several features are correlated as the Gini index tends to dilute over different features in different trees. To evaluate an individual feature's importance, we used the positive predictive value (PPV) from single decision tree classifiers trained on individual Cell Painting features. These classifiers were trained on 486 compounds (85 mitotoxic) having max depth of one and two leaf nodes on our dataset for each feature. The tree hence determines an optimal threshold per feature to distinguish mitochondrial toxic compounds from non-toxic compounds. The mean PPV of Cell Painting features having PPV > 0 was grouped by compartment (Cells, Cytoplasm, and Nuclei), channel (AGP, Nucleus, ER, Mito, Nucleolus/Cytoplasmic RNA), and feature group (Correlation, Granularity, Intensity, Radial Distribution, Texture). The predictive value of individual Gene Expression features was computed in a similar manner using decision tree classifiers on 768 compounds (161 mitotoxic).

**Types of features' combinations used**. Here we employed 5 types of models having different input features, combinations thereof as well as model ensembling. Initially, Cell Painting features, Gene Expression features and Morgan fingerprints were used separately as features for three separate models. As shown in Fig. S11, an early-stage model fused Cell Painting, Gene Expression and Morgan fingerprints by appending the features into a single vector while another late-stage fusion model averaged the probabilities of the three models using only Cell Painting, Gene Expression and Morgan fingerprints respectively into a single probability value.

**Model generation and evaluation**. 382 compounds (62 mitotoxic) from the mitochondrial toxicity data having both Cell Painting and Gene Expression annotations were used for modelling. Given the size of the training data, an artificial neural network model cannot capture the inherent data distribution effectively to perform well in an external test set (see Supplementary Data 10 for modelling results when comparing different Random Forests and artificial neural

network model). Hence, in this study, Random Forest models were trained using scikit-learn[103].

As shown in Fig. S12, we used a grid search with balanced accuracy as the scoring function as implemented in scikit-learn, a Python based package, to optimise hyperparameters. Using a 4-fold stratified cross validation we determined the variation in performance when changing number of trees in the Random Forest. We used a GridSearchCV. The grid parameters varied the number of trees from 21 to 301 with a step size of 5. We checked for change in balanced accuracy in out of fold results in the cross validation and training time. No considerable improvement was observed on increasing number of trees. Hence, we opted for the optimal baseline model with 100 trees which is within the optimal range of 64−128 number of trees given in previous research[79] has also shown that increasing the number of trees does not necessarily improve performance[104]. The nodes were let to expand until all leaves were pure or until all leaves contained less than a minimum of 2 samples that are required to split an internal node. A minimum of 1 sample was required to be at a leaf node and the number of features to consider when looking for the best split was set as the square root of total features. The consistent performance is most likely as Random Forests are usually robust against overfitting. As shown in Fig. S13, we used 4-fold nested cross-validation; inside the outer loop, a 4-fold stratified splitting divided the data into a training set (75%), on which feature selection was performed and the remaining into a test set (25%). Inside the inner loop, a Random Forest model with parameters as above was trained on the training set using 4-fold stratified cross-validation. For each model, to account for class imbalance, we tuned the threshold of probability to determine the cut-off for toxicity labels having maximum value for Youden's J statistic (J = True Positive Rate – False Positive Rate). The Youden index is frequently used to detect an optimal threshold to be used as a criterion for classifying subjects without biasing the model towards one class. Thus, the predictions can be used to fully exploit the model giving equal weights to sensitivity and specificity without favouring one of them. From combined results of the out-of-fold data from cross-validation, we chose the threshold of probability with the largest Youden's J statistic value. This threshold was then used for the test set (hence the test set was not used directly while selecting the optimal threshold). The entire process of nested cross-validating was repeated 50 times; we evaluated our models on the distribution and median of the performance metrics from all 200 test sets. The models overall trained with reasonable training time and threshold balancing ensured that overfitting on an unbalanced dataset could be avoided.

**External model evaluation**. For the external test set, we trained 5 Random Forest models for each feature/combination on our dataset using 4-fold cross-validation and the optimal threshold was determined similarly from the combined out-of-fold data. The model was then retrained on the entire dataset and used to predict the external test set with the threshold previously determined.

**Evaluation metrics**. F1 scores of the minority class (mitotoxic compounds), precision of the minority class (mitotoxic compounds), sensitivity, specificity, Balanced Accuracy (BA), Area Under Curve-Receiver Operating Characteristic (AUC-ROC), Area Under Curve-Precision Recall (AUCPR), and Mathew's correlation constant (MCC) were used to assess model performance as implemented in scikit-learn python package[103]. Often in a toxicity prediction problem with unbalanced data, the number of nontoxic compounds far outweighs the number of mitotoxic compounds and improvement in the prediction of the mitotoxic compounds (minority class) is desired[105]. Here particular metrics such as sensitivity and AUCPR are useful and less likely to exaggerate model performance. For comparing model predictions to true values in the external test set, F1 scores and precision of the minority class and the sensitivity of the model were used as they focus on the minority class (mitotoxic compounds) being detected by the model.

**Extrapolation to new structural/morphological space**. To evaluate if our models can extrapolate to novel chemical space (either in structural space or in morphological space) we defined for each compound in the external test set two parameters: (1) Structural distance to the training set:

$$Structural\ distance = median\left(x_1, x_2 \ldots, x_5\right) \qquad (i)$$

where $x_k$ = pairwise Tanimoto distances in decreasing magnitude, where,

$$Tanimoto\ distance = 1 - Tanimoto\ similarity \qquad (ii)$$

and (2) Morphological distance to the training set:

$$Morphological\ distance = 1 - Morphological\ similarity \qquad (iii)$$

where,

$$Morphological\ similarity = abs\left(median\left(a_1, a_2 \ldots a_{15}, b_1, b_2 \ldots b_{15}\right)\right) \qquad (iv)$$

and $a_k$ = positively and $b_k$ = negatively pairwise Pearson correlations in decreasing magnitude.

The distances were defined the same as the subsection *"Comparing class separation and visualization of compounds in morphological space"*. The structural distance was defined as the median of the five lowest Tanimoto distances[106] between Morgan fingerprints of the test compound and the compounds in the

training dataset of the same activity annotation. The morphological distance was defined as the one minus the absolute value of the median of 15 most positively and 15 most negatively pairwise Pearson correlations (using selected Cell Painting features) of the test compound and the compounds in the training dataset of the same activity annotation. In this manner, we could evaluate if true positives from test sets for each model lie in relatively distant structural or morphological space to their training space.

**Statistics and Reproducibility**. A detailed description of each analysis' steps and statistics is contained in the methods section of the paper. Statistical methods were implemented using the pandas Python package[102]. Machine learning models, hyperparameter optimisation and evaluation metrics were implemented using scikit-learn, a Python based package[103]. The sample numbers *n* for each analysis are listed in the figure captions. We released the code and training and external test set data for the models publicly at https://git.io/JDGyc and in the Supplementary Data.

**Reporting summary**. Further information on research design is available in the Nature Research Reporting Summary linked to this article.

## Data availability
The training dataset (used for nested cross validation) and the external test set used in this study are released in Supplementary Data 11 and 12 and other data are publicly available at https://git.io/JDGyc. Any queries regarding data can be addressed to the corresponding author.

## Code availability
We released the python code for our models which are publicly available at https://git.io/JDGyc.

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

## Acknowledgements

S.S. acknowledges the Cambridge Commonwealth, European and International Trust, Boak Student Support Fund (Clare Hall), Jawaharlal Nehru Memorial Fund, Allen, Meek and Read Fund, and Trinity Henry Barlow (Trinity College) for providing funding for this study. S.S. gratefully acknowledge discussions with Prof. Michele Vendruscolo (University of Cambridge). OS acknowledges funding from Swedish Research Council under grant #2020-01865 and #2020-03731. This work was performed using resources provided by the Cambridge Service for Data Driven Discovery (CSD3) operated by the University of Cambridge Research Computing Service (www.csd3.cam.ac.uk), provided by Dell EMC and Intel using Tier-2 funding from the Engineering and Physical Sciences Research Council (capital grant EP/P020259/1), and DiRAC funding from the Science and Technology Facilities Council (www.dirac.ac.uk). Cartoons in Figs. 1, 2, 4–7 were created with Bioicons (https://bioicons.com) which compiled images from the Database Centre for the life sciences/TogoTV (© 2016 DBCLS TogoTV, https://togotv.dbcls.jp/en/pics.html) and Servier (https://smart.servier.com).

## Author contributions

All authors have given approval to the final version of the manuscript. S.S. designed and performed exploratory data analysis, implemented, and trained the models. S.S, J.C.P and O.S analysed the biological interpretation of morphological features. S.S, M.A.T and H.Y contributed to analysing the results of models. S.S. wrote the manuscript with extensive discussions with O.S and J.C.P. A.B. and O.S. supervised the project. All the authors (S.S., J.C.P, M.A.T, H.Y, O.S and A.B) reviewed, edited, and contributed to discussions on the manuscript.

## Competing interests

The authors declare no competing interests.
