## [Peer Review File · Communications Biology]

Reviewers' comments:

Reviewer #1 (Remarks to the Author):

In the manuscript, the author integrated three types of data, Cell Painting(CP), Gene Expression(GE) and structure features, to predict mitochondrial toxicity based on random forest models. Compared with using any of the three features alone, the fusion models using the three features together has a certain improvement in the prediction performance on the external validation set. The results proved to a certain extent that adding CP and GE features to the models improved the generalization ability of the models. The work illustrated the significance of CP and GE features, especially CP features, in predicting mitochondrial toxicity. Therefore, It is recommend to accept the paper.

Here are some suggestions:

- (1) Some descriptions are inconsistent. For example, in the section "Extrapolation to new structural/morphological space" in Methods, the formula of structure distance (the number of samples is 15) does not match the description in the text (The number of samples is 5). The subsection "Statistical Analysis of Cell Painting features and Gene Expression features" mentioned below is not found elsewhere in the paper.
- (2) The pictures containing chemical structures is not clear enough. Some descriptions cannot easily find corresponding subgraphs. It is friendly for readers to use higher-resolution pictures and accurate description of referenced subgraphs.
- (3) It is suggested that the author further optimize the English expression. Excessive use of attributive clauses seems to result in overly long sentences and problems with ambiguous referential meaning. In addition, the grammar of some sentences is not standardized, for example, inappropriate or multiple use of the preposition or conjunction in a sentence.

Reviewer #2 (Remarks to the Author):

The authours proposed an interesting work on the detection of mitochondrial toxicants by combining chemical descriptors with different levels of biological readouts. The manuscript is well written, so I suggest it be published after minor revision.

Detailed points:

1. There are many features used in this study, but the data set is a liitle small, so how to avoid the problem of over-fitting?
2. Due to the need to use a lot of features, the training set that can be used is relatively small. If a new compound is used, will the result be worse due to the change of chemical space?
3. In Morphology data, the changes of images may be particularly obvious due to cytotoxicity and other reasons. At this time, it may not be guaranteed that there is indeed mitochondrial toxicity. Has the author considered this situation.

Reviewer #3 (Remarks to the Author):

This manuscript uses the random forest model to predict mitochondrial toxicity from hypothesis-free data, including L1000 gene expression data, Cell Painting microscopy data, and chemical structure information from Morgan fingerprints for 382 chemicals tested in the Tox21 mitochondrial membrane potential assay. This is an essential topic since mitochondrial toxicity has been connected to drug side effects; however, the underlying mechanisms and how to predict mitochondrial toxicity are still under investigation. Overall, the paper contributes significantly to the field and is worth publishing in Communications Biology. It is also very helpful that the authors make the related computer code and data available on GitHub. However, some concerns and issues need to be resolved before being accepted and published.

Major concerns:

1. Although the authors combined Cell Painting, gene expression features, and Morgan fingerprints to improve model performance on an external test set of 236 compounds by 60% in terms of F1 score and improved extrapolation to new chemical space, the overall performance of the model is still not very good. The F1 scores and MCCs listed in tables S1 and S6 are still low (mostly <0.5),

and the balanced accuracy is 0.68 (<0.7). Have the authors tried using different types of machine learning or deep learning models, such as neural networks, gradient boosting, or a variational autoencoder, to improve the model's predictions?

2. The data used in this manuscript came from different cell types. Tox21 used HepG2 as testing cell lines, Cell Painting used USO2 cells, and LINC L1000 used a variety of cell lines (MCF7, A549, HepG2, HT29, etc.). How did the authors solve the issues of toxicity discrepancy in assays performed in different tissues and perturbants at different concentrations?

3. Is it possible to increase the number of compounds used in the testing sets using literature (as in table S1) or through manual curation? There are also publicly available databases (for example, the MitoTox database) that record compounds that show mitochondrial toxicity under different assays and dosages.

4. In the case of predicting a result from an inconclusive, besides providing literature support, is it possible to perform additional experiments to validate the model predictions?

5. How widely could the random forest model proposed in this manuscript be used? The combined model required both gene expression and morphology data. Could the model easily be applied to new chemicals that don't have those gene expressions and morphology data (for example, new drugs or compounds synthesized by pharmaceutical companies)?

Minor concerns:

1. The authors used the term "inconclusive compounds" for those compounds that had inconclusive assay outcomes in Tox21 due to excessive cytotoxicity. It will be helpful to define what "inconclusive compounds" are in the manuscript.

2. Figure 1 seems oversimplified. There are multiple factors affecting mitochondrial toxicity. Besides ER stress, oxidative stress, proteotoxic stress, and apoptosis, there are other types of signaling and outcomes (though cell death might be the ultimate outcome when cytotoxicity is serious). Unfolded protein response was mentioned in the manuscript but is not shown in Figure 1. The authors should also consider retrograde signaling (cross-talk between mitochondria and nucleus), autophagy or mitophagy, mitochondrial-derived vesicles, MVB, etc. It is good that the manuscript discussed some intraorganelle communications related to mitochondrial toxicity in organelle and cellular morphology and gene expression in Figures 5, 6, and 7.

3. In Figure 4, are there any other compounds that have the same chemical structure category that are not in the mitochondrial toxicity clusters? For example, not all statins show mitochondrial toxicity, and not all statins have the same mechanisms of mitochondrial toxicity.

4. In tables S11 and S12, it would be helpful to list all the compounds' common names and IUPAC names for the training data and external test data.

5. There is no mention or discussion of existing literature related to combining Cell Painting data and gene expression data. For example, Haghighi et al. (bioRxiv 2021, <https://doi.org/10.1101/2021.09.08.459417>) and Nassiri et al. (Nucleic Acids Research, Volume 46, Issue 19, 2 November 2018, page e116, <https://doi.org/10.1093/nar/gky626>).

6. Please provide more details on the results of hyperparameter tuning and cross-validation in addition to those in Figures S12 and S13.

Respected Editor,

Thank you so much for the reviews which we were happy to implement as outlined below.

We hope that the revised version of this work is suitable for publication in your journal.

Thanking you

Yours sincerely

Srijit

Reviewers' comments:

Reviewer #1 (Remarks to the Author):

In the manuscript, the author integrated three types of data, Cell Painting(CP), Gene Expression(GE) and structure features, to predict mitochondrial toxicity based on random forest models. Compared with using any of the three features alone, the fusion models using the three features together has a certain improvement in the prediction performance on the external validation set. The results proved to a certain extent that adding CP and GE features to the models improved the generalization ability of the models. The work illustrated the significance of CP and GE features, especially CP features, in predicting mitochondrial toxicity. Therefore, It is recommend to accept the paper.

Here are some suggestions:

(1) Some descriptions are inconsistent. For example, in the section "Extrapolation to new structural/morphological space" in Methods, the formula of structure distance (the number of samples is 15) does not match the description in the text (The number of samples is 5). The subsection "Statistical Analysis of Cell Painting features and Gene Expression features" mentioned below is not found elsewhere in the paper.

Thank you for your comments, we have now edited the section on p.32 "subsection 'Comparing class separation and visualization of compounds in morphological space'. We apologise for the error on number of samples in the formula, it is indeed 5 and we have now corrected this in the formula on p.31.

(2) The pictures containing chemical structures is not clear enough. Some descriptions cannot easily find corresponding subgraphs. It is friendly for readers to use higher-resolution pictures and accurate description of referenced subgraphs.

We have now updated figure 9 with high resolution chemical structures on p.58. We have also enlarged chemical structures on Figure 4 (p.52) and double-checked cluster numbering, while naming compounds on p.9-10 to ensure the reader is able to locate the compounds in the figures easily.

(3) It is suggested that the author further optimize the English expression. Excessive use of attributive clauses seems to result in overly long sentences and problems with ambiguous referential meaning. In addition, the grammar of some sentences is not standardized, for example, inappropriate or multiple use of the preposition or conjunction in a sentence.

Thank you for the comment, we have now revised the manuscript to correct multiple places where wrong prepositions were used and checked language throughout.

Reviewer #2 (Remarks to the Author):

The authors proposed an interesting work on the detection of mitochondrial toxicants by combining chemical descriptors with different levels of biological readouts. The manuscript is well written, so I suggest it be published after minor revision.

Detailed points:

1. There are many features used in this study, but the data set is a little small, so how to avoid the problem of over-fitting?

Thank you for your comments and for reading through the manuscript.

We agree with the reviewer that with ensemble methods like Random Forests we need to be aware of overfitting. We avoid overfitting first using a feature selection as described in p.26

“While the Random Forest algorithm employed for modelling (see Model generation and evaluation below) is in principle able to select features, this is not always successful which made us compare different explicit feature selection methods in parallel with inputting all features into the models subsequently. This method in our experience led to less overfitting while still being interpretable and able to extrapolate to the external test set compared to other methods”

Second, we used a parameter optimisation (as described on p.29)

“..., we used a grid search with balanced accuracy as the scoring function as implemented in scikit-learn, a Python based package, to optimise hyperparameters. Using a 4-fold stratified cross validation we determined the variation in performance when changing number of trees in the Random Forest. We used a GridSearchCV. The grid parameters varied the number of trees from 21 to 301 with a step size of 5. We checked for change in balanced accuracy in out of fold results in the cross validation and training time. No significant improvement was observed on increasing number of trees. Hence, we opted for the optimal baseline model with 100 trees which is within the optimal range of 64–128 number of trees given in previous research [77] has also shown that increasing the number of trees does not necessarily improve performance.[103]”

[77] Oshiro, T. M., Perez, P. S. & Baranauskas, J. A. How many trees in a random forest? Lect. Notes Comput. Sci. (including Subser. Lect. Notes Artif. Intell. Lect. Notes Bioinformatics) 7376 LNAI, 154–168 (2012)

[103] Jain, S., Kotsampasakou, E. & Ecker, G. F. Comparing the performance of meta-classifiers—a case study on selected imbalanced data sets relevant for prediction of liver toxicity. J. Comput. Aided. Mol. Des. 32, 583–590 (2018).

We have now also updated the manuscript with a new section on the “Limitations and Future Work” on p.18

To account for overfitting : “One limitation of the here presented study is the size of the dataset (overlap between the Cell Painting and Gene Expression features with mitochondrial toxicity assay annotations), making it cover a somewhat limited area of chemical space. To mitigate the risk of overfitting in an ensemble method such as Random Forests, first the Cell Painting and Gene Expression features were subjected to feature selection. We used hyperparameter optimisation and nested cross validation to evaluate the training data and selected the optimal number of trees as shown in previous work. [77] Although an individual decision tree may be more prone to over-fitting, in the case of an ensemble method such as Random Forests, we also avoid overfitting by bootstrapping samples (randomly choosing selected features at each split for trees).[78] To further examine this, we used an external test set where we found the performance does drop in the external test set when using only chemical structure but not when using fusion models combining structure

with cell morphology and gene expression. Our evaluation and discussions on extrapolation to new chemical space (as shown in Figure 9) are based on this external test set and hence are still valid.”

[77] Oshiro, T. M., Perez, P. S. & Baranauskas, J. A. How many trees in a random forest? Lect. Notes Comput. Sci. (including Subser. Lect. Notes Artif. Intell. Lect. Notes Bioinformatics) 7376 LNAI, 154–168 (2012)

[78] Breiman, L. Bagging predictors. Mach. Learn. 24, 123–140 (1996).

2. Due to the need to use a lot of features, the training set that can be used is relatively small. If a new compound is used, will the result be worse due to the change of chemical space?

Thank you for your comments we have now updated the discussions on p.20 to include:

“We show that the late-stage fusion model has higher accuracy when predicting mitochondrial toxicity (F1 score 0.42) when extrapolating to the new chemical space of an external set (wherein compounds which were structurally dissimilar to the training set as shown in Supplementary Figure S6) compared to the model using only Morgan fingerprints (F1 score of 0.25). As shown in Figure 9, the utilization of cell morphology and gene expression data improved the detection of mitotoxic compounds (as shown in late-stage fusion model with sensitivity 0.79 vs 0.19 when using only structural data). Hence, we can conclude that the detection of mitochondrial toxicity is improved when using all three feature spaces (structure, morphology, and gene expression) together.”

And further on p.18

“...To further examine this, we used an external test set where we found the performance does drop in the external test set when using only chemical structure but not when using fusion models combining structure with cell morphology and gene expression. Our evaluation and discussions on extrapolation to new chemical space (as shown in Figure 9) are based on this external test set and hence are still valid.”

Supplementary Figure S6: Pairwise combinations of training (382 compounds) vs test set (244 compounds) having highest Tanimoto similarity (Tc) using (a) Morgan Fingerprints of radius 2 and 2048 bits and (b) 166

public MACCS keys. (c) and (d) show clustered heatmaps of all pairwise Tanimoto similarity coefficients using Morgan Fingerprints of radius 2 and 2048 bits and 166 public MACCS keys respectively. This shows most test compounds are structurally diverse to the training set (with $T_c < 0.40$ for Morgan Fingerprints and $T_c < 0.85$ for MACCS keys).

3. In Morphology data, the changes of images may be particularly obvious due to cytotoxicity and other reasons. At this time, it may not be guaranteed that there is indeed mitochondrial toxicity. Has the author considered this situation.

We edited the Methods Section p.24 to explain how we removed compounds causing cell death in the Cell Painting dataset

“Further, the changes in morphology may be particularly obvious due to excessive cytotoxicity there we must avoid perturbations which drastically reduced the cell count compared to the neutral control of DMSO. We removed such compounds (step 1, Supplementary Figure S9a) in Cell Painting images by removing compounds with a threshold of 1.5 times standard deviation below the mean of “Cells Number Object Number” (the distribution is shown in Supplementary Figure S9b) which is below -15.09.”

We have now updated the manuscript with new Figure S9(b)

(b) The distribution of Cell Count among the 542 compounds for which Cell Painting and Tox21 Mitochondrial toxicity annotations were available. We removed compounds which drastically reduced the cell count compared to the neutral control of DMSO (by a threshold of 1.5 times standard deviation below the mean of “Cells Number Object Number” which is below -15.09).

And edited Methods p.22 to explain how we removed compounds causing cell death in the Tox21 dataset

“The compounds causing excessive cytotoxicity were labelled “inconclusive” which helps differentiate compounds that decreased MMP from those inducing high cytotoxicity. One cannot be certain if mitochondrial dysfunction may have caused the excessive cell death in these “inconclusive compounds”. Hence as a precautionary measure to not end with predicting cytotoxicity in this study, but only mitochondrial toxicity, we removed inconclusive compounds from the training dataset. Hence, in our models, inconclusive compounds were removed and for the remaining compounds, mitochondrial toxicity labels were assigned as per assay hit calls from the Tox21 summary assay.”

Reviewer #3 (Remarks to the Author):

This manuscript uses the random forest model to predict mitochondrial toxicity from hypothesis-free data, including L1000 gene expression data, Cell Painting microscopy data, and chemical structure information from Morgan fingerprints for 382 chemicals tested in the Tox21 mitochondrial membrane potential assay. This is an essential topic since mitochondrial toxicity has been connected to drug side effects; however, the underlying mechanisms and how to predict mitochondrial toxicity are still under investigation. Overall, the paper contributes significantly to the field and is worth publishing in *Communications Biology*. It is also very helpful that the authors make the related computer code and data available on GitHub. However, some concerns and issues need to be resolved before being accepted and published.

Major concerns:

1. Although the authors combined Cell Painting, gene expression features, and Morgan fingerprints to improve model performance on an external test set of 236 compounds by 60% in terms of F1 score and improved extrapolation to new chemical space, the overall performance of the model is still not very good. The F1 scores and MCCs listed in tables S1 and S6 are still low (mostly <0.5), and the balanced accuracy is 0.68 (<0.7). Have the authors tried using different types of machine learning or deep learning models, such as neural networks, gradient boosting, or a variational autoencoder, to improve the model's predictions?

We thank the author for their comments.

We have now included results from training a multilayer perceptron (MLP) is a feedforward artificial neural network model on the external test in Supplementary Table S10 and included a comment about the same on p.29

“Given the size of the training data, an artificial neural network model cannot capture the inherent data distribution effectively to perform well in an external test set (see Supplementary Table S10 for modelling results when comparing different Random Forests and artificial neural network model).”

The key limitation of this study, as now emphasized in the Limitations and Future Work section (p.18) is the underlying data; and at the same time, we wanted to have interpretable models. Given its size, deep learning models or neural networks are not suitable; in addition, the method is not interpretable. We have now included this on p19:

“Future studies would benefit from larger datasets, such as the announced future data depositions from the JUMP consortium [80], and also more and better annotated compounds that show mitochondrial toxicity under different assays and dosages such as from the Mitotox database[46]. It may also be possible to apply different types of machine learning or deep learning models, such as deep neural networks, gradient boosting, or a variational autoencoder (which has been previously shown to reveal an interpretable latent space[81]) to improve the model's predictions and generally improve the interpretability of models. ”

[80] JUMP-Cell Painting Consortium. <https://jump-cellpainting.broadinstitute.org/>.(accessed May 2, 2022)

[46] Lin, Y. Te, Lin, K. H., Huang, C. J. & Wei, A. C. MitoTox: a comprehensive mitochondrial toxicity database. *BMC Bioinform* 22, 1–14 (2021).

[81] Chow, Y. L., Singh, S., Carpenter, A. E. & Way, G. P. Predicting drug polypharmacology from cell morphology readouts using variational autoencoder latent space arithmetic. *PLOS Comput. Biol.* 18, e1009888 (2022).

2. The data used in this manuscript came from different cell types. Tox21 used HepG2 as testing cell lines, Cell Painting used USO2 cells, and LINC L1000 used a variety of cell lines (MCF7, A549, HepG2, HT29, etc.).

How did the authors solve the issues of toxicity discrepancy in assays performed in different tissues and perturbants at different concentrations?

Thank you for the comments, we have now updated the Limitations and Future work section to include (p.19):

“Another limitation is the discrepancy between the cell lines, where Cell Painting was carried out on USO2 cells, and LINC L1000 used a variety of cell lines (MCF7, A549, HepG2, HT29, etc.); we use the Cell Painting to predict toxicity in the Tox21 assay which used HepG2 as testing cell lines thus giving rise to a toxicity discrepancy in assays using different tissues and perturbants at different concentrations. In our case, we leverage the fact that cell morphological data provides versatile biological data that is generally able to extrapolate to different cell lines as shown in previous studies. For example, Cox et. al. used cell morphological data from 15 reporter cell lines to predict the mechanism of action (MOA) but did not see any individual reporter cell line outperform others (with the notable exception of GR agonists).[79] They also observed that the genetic background of the reporter cell line did not affect the overall AUC-ROC values calculated for the different MOAs. Although gene expression data can be cell line specific, previous work by Lapins et al showed that the prediction of MOA was similarly effective with an average AUC of 0.83 across 3 different cell types.[34] Therefore as outlined in the Methods Section Gene Expression features subsection, for gene expression features in this study we used the strongest signatures irrespective of the cell line. In this manner, although the cell line of feature spaces differs from the toxicity assay (and more so in the case of organ-level toxicity), we could leverage the biological information in the cell morphology and gene expression data to predict mitochondrial toxicity which is an in vitro toxicity endpoint.”

[79] Cox, M. J. et al. Tales of 1,008 small molecules: phenomic profiling through live-cell imaging in a panel of reporter cell lines. *Sci. Rep.* 10, 1–14 (2020).

[34] Lapins, M. & Spjuth, O. Evaluation of Gene Expression and Phenotypic Profiling Data as Quantitative Features for Predicting Drug Targets and Mechanisms of Action. *bioRxiv* 580654 (2019) doi:10.1101/580654.

3. Is it possible to increase the number of compounds used in the testing sets using literature (as in table S1) or through manual curation? There are also publicly available databases (for example, the MitoTox database) that record compounds that show mitochondrial toxicity under different assays and dosages.

Thank you for your comment. The Mitotox dataset was released towards the end of conducting this study. We have now updated our manuscript to include compounds from this database.

We looked at the Mitotox database for compounds associated to decrease transmembrane potential (<https://www.mitotox.org/functions/4717/detail/>). After pre-processing and standardisation of InChI we obtained 652 unique compounds. Out of these, both Cell Painting and Gene Expression data was available for only 71 compounds and 63 out of these 71 compounds were already used in this study (54 were used in the training data and 9 in the external test set) We have now added the remaining 8 compounds to our external test dataset and updated the manuscript.

Methods p.23

“We further included compounds from Mitotox Database[46] compiling mitochondrial toxicity under different assays and dosages. We searched Mitotox for compounds associated to decreased transmembrane potential [86] to obtain 652 unique mitotoxic compounds.”

[86] MITOTOX. <https://www.mitotox.org/functions/4717/detail/>.

[46] Lin, Y. Te, Lin, K. H., Huang, C. J. & Wei, A. C. MitoTox: a comprehensive mitochondrial toxicity database. *BMC Bioinform.* 22, 1–14 (2021).

Methods p.25

“From the Mitotox database we obtained 652 unique compounds, out of which both Cell Painting and Gene Expression data were available for only 71 compounds. 63 out of these 71 compounds were already used in

this study (54 were used in the training data and 9 in the external test set) We added the remaining 8 compounds to our external test dataset thus totalling 244 distinct compounds (47 mitotoxic and remaining nontoxic). To evaluate our models, we used this an external test set where compounds in this external test set do not appear in the training data.”

All relevant figures and tables were updated with new results. Mitotox is a smaller dataset with 1400 compounds compared to the original Tox21 dataset with more than 7000 unique compounds. The authors combined Tox21 mitochondrial membrane potential assay results which is the same source data we use. The size of the dataset used in this study even after including MitoTox, is far more limited from the availability of public domain feature space of the Cell Painting assay and Gene Expression data. With the availability of the new dataset is released from Cell Painting JUMP we hope to be able to increase the size of the dataset for future studies.

This is now outlined in the Limitations section:

“Future studies would benefit from larger datasets, such as the announced future data depositions from the JUMP consortium [80], and also more and better annotated compounds that show mitochondrial toxicity under different assays and dosages such as from the Mitotox database[46].”

[80] JUMP-Cell Painting Consortium. <https://jump-cellpainting.broadinstitute.org/>.(accessed May 2, 2022)

[46] Lin, Y. Te, Lin, K. H., Huang, C. J. & Wei, A. C. MitoTox: a comprehensive mitochondrial toxicity database. BMC Bioinform 22, 1–14 (2021).

4. In the case of predicting a result from an inconclusive, besides providing literature support, is it possible to perform additional experiments to validate the model predictions?

Given the computational nature of our lab we could not perform wet lab experiments for validation as part of this study. Hence, we looked at literature support for different conditions/cell lines etc. to determine if such compounds cause mitochondrial toxicity.

5. How widely could the random forest model proposed in this manuscript be used? The combined model required both gene expression and morphology data. Could the model easily be applied to new chemicals that don't have those gene expressions and morphology data (for example, new drugs or compounds synthesized by pharmaceutical companies)?

It is true that we will need both gene expression and morphology data for the models to be applied; however, given that those readouts have potential relevance for both efficacy- and safety-related endpoints this would still be attractive in an industrial drug discovery setting.

We have now added the following text to the Discussion on p.21

“The Cell Painting feature space is less expensive than the L1000 assay or DRUG-seq[82] and thus enables larger high-throughput experiments.[40] This cell morphology modality is thus being increasingly explored both in the public domain such as by the JUMP consortium[80]and as well as by pharmaceutical companies such as by Janssen[83]. Given that Cell Painting readouts can be used for multiple purposes, this supports their use also for the prediction of a mitochondrial toxicity endpoint.”

[80] JUMP-Cell Painting Consortium. <https://jump-cellpainting.broadinstitute.org/>.(accessed May 2, 2022)

[82] Ye, C. et al. DRUG-seq for miniaturized high-throughput transcriptome profiling in drug discovery. Nat. Commun. 9, 4307 (2018).

[40] Way, G. P. et al. Morphology and gene expression profiling provide complementary information for mapping cell state. bioRxiv 2021.10.21.465335 (2021)

[83] De Wolf, H. et al. High-Throughput Gene Expression Profiles to Define Drug Similarity and Predict Compound Activity. *Assay Drug Dev. Technol.* 16, 162–176 (2018).

Cell Painting and Gene Expression data will make these models more reliable in prediction outside of structural domain and thus save time and cost if they are predictive of more complex assays. We have now added a statement in this regard to the Discussion p.20-21

“We show that the late-stage fusion model has higher accuracy when predicting mitochondrial toxicity (F1 score 0.42) when extrapolating to the new chemical space of an external set (wherein compounds which were structurally dissimilar to the training set as shown in Supplementary Figure S6) compared to the model using only Morgan fingerprints (F1 score of 0.25). As shown in Figure 9, the utilization of cell morphology and gene expression data improved the detection of mitotoxic compounds (as shown in late-stage fusion model with sensitivity 0.79 vs 0.19 when using only structural data). Hence, we can conclude that the detection of mitochondrial toxicity is improved when using all three feature spaces (structure, morphology, and gene expression) together.”

Minor concerns:

1. The authors used the term “inconclusive compounds” for those compounds that had inconclusive assay outcomes in Tox21 due to excessive cytotoxicity. It will be helpful to define what “inconclusive compounds” are in the manuscript.

We have now refined this in the Results section on p.15

“Next, we compared predictions using the 5 models above for the 22 compounds from the data where inconclusive results were obtained in the Tox21 due to excessive cytotoxicity either in the mitochondrial depolarization assay or in the cell viability assay. [69]”

[69] Attene-Ramos, M. S. et al. Profiling of the Tox21 Chemical Collection for Mitochondrial Function to Identify Compounds that Acutely Decrease Mitochondrial Membrane Potential. *Environ. Health Perspect.* 123, 49–56 (2015).

Further, we also updated the text on p.22 for the Methods Section

“Thus, the summary assay not only considers triplicate runs of the ratio (red/green) readout in the MMP assay but also each fluorescence channel separately, as well as the cytotoxicity results.[69] The compounds causing excessive cytotoxicity were labelled “inconclusive” which helps differentiate compounds that decreased MMP from those inducing high cytotoxicity. One cannot be certain if mitochondrial dysfunction may have caused the excessive cell death in these “inconclusive compounds”. Hence as a precautionary measure to not end with predicting cytotoxicity in this study, but only mitochondrial toxicity, we removed inconclusive compounds from the training dataset. Hence, in our models, inconclusive compounds were removed and for the remaining compounds, mitochondrial toxicity labels were assigned as per assay hit calls from the Tox21 summary assay.”

[69] Attene-Ramos, M. S. et al. Profiling of the Tox21 Chemical Collection for Mitochondrial Function to Identify Compounds that Acutely Decrease Mitochondrial Membrane Potential. *Environ. Health Perspect.* 123, 49–56 (2015).

2. Figure 1 seems oversimplified. There are multiple factors affecting mitochondrial toxicity. Besides ER stress, oxidative stress, proteotoxic stress, and apoptosis, there are other types of signaling and outcomes (though cell death might be the ultimate outcome when cytotoxicity is serious). Unfolded protein response was mentioned in the manuscript but is not shown in Figure 1. The authors should also consider retrograde signaling (cross-talk between mitochondria and nucleus), autophagy or mitophagy, mitochondrial-derived vesicles, MVB, etc. It is good that the manuscript discussed some intraorganellar communications related to mitochondrial toxicity in organelle and cellular morphology and gene expression in Figures 5, 6, and 7.

We have now updated Figure 1 to include Unfolded protein response. We have also added a paragraph in the introduction on the multiple factors affecting mitochondrial toxicity to discuss some of the signalling and outcomes discussed by the referee in p.4

“There are multiple factors affecting mitochondrial toxicity. A common direct cause of mitochondrial dysfunction is uncoupling of the electron transport chain from ATP synthesis or accumulation of calcium in mitochondria causing an increase in Reactive Oxygen Species (ROS), leading to oxidative stress and damaging mitochondrial DNA (mtDNA). [7] Indirect effects of drugs on cells such as inhibition of fatty acid β -oxidation, uncoupling of oxidative phosphorylation, the opening of the membrane permeability transition pore, and disruption of mtDNA synthesis and translation have also been shown to cause mitochondrial toxicity. [7] Retrograde signalling pathways often triggered by these mechanisms result in cross-talk between mitochondria and nucleus leading to changes in nuclear gene expression and may activate unfolded protein response. [8] Besides ER stress, oxidative stress, proteotoxic stress, and apoptosis, there are also other types of signalling and outcomes in aspects of mitochondrial organelle biology such as mitophagy and mitochondrial-derived vesicles [9] that may contribute to mitochondrial toxicity although the outcome may be cell death when there is executive cytotoxicity. Hence we can see that mitochondrial toxicity can be difficult to predict with just chemical structure and there is a need to include more biological data which may be more predictive of this endpoint.”

[7] Chan, K., Truong, D., Shangari, N. & O'Brien, P. J. Drug-induced mitochondrial toxicity. *Expert. Opin. Drug. Metab. Toxicol.* 1, 655–669 (2005).

[8] Granat, L., Hunt, R. J. & Bateman, J. M. Mitochondrial retrograde signalling in neurological disease. *Philos. Trans. R. Soc. B Biol. Sci.* 375, (2020).

[9] Cadete, V. J. J. et al. Formation of mitochondrial-derived vesicles is an active and physiologically relevant mitochondrial quality control process in the cardiac system. *J. Physiol.* 594, 5343–5362 (2016).

3. In Figure 4, are there any other compounds that have the same chemical structure category that are not in the mitochondrial toxicity clusters? For example, not all statins show mitochondrial toxicity, and not all statins have the same mechanisms of mitochondrial toxicity.

We have now added in the manuscript the following on p.10 under subsection Cell Painting features cluster mitochondrial toxicants to identify different mechanisms of mitochondrial toxicity

“Although compounds in the individual clusters were often structurally dissimilar to each other, we did not find any other compound in the training dataset (542 compounds) with a very similar chemical structure (greater than 0.85 Tanimoto similarity) to the compounds in the individual mitochondrial toxicity clusters. This shows morphology space could cluster dissimilar structures with similar modes of action together and did not miss similar compounds with similar modes of action.”

4. In tables S11 and S12, it would be helpful to list all the compounds' common names and IUPAC names for the training data and external test data.

We have now added these to the Supplementary Information Table S11 and S12.

5. There is no mention or discussion of existing literature related to combining Cell Painting data and gene expression data. For example, Haghghi et al. (bioRxiv 2021, <https://doi.org/10.1101/2021.09.08.459417>) and Nassiri et al. (Nucleic Acids Research, Volume 46, Issue 19, 2 November 2018, page e116, <https://doi.org/10.1093/nar/gky626>).

We have now added a sentence in the Introduction discussing the biological information content in both feature spaces on p.6

“Further, it has been also shown that such cell morphology space provides a feature-specific subspace that is complementary to biological information contained in gene expression [39], which has also been shown in an application-based problem in predicting the mechanism of action of compounds. [40]”

[39] Haghghi, M., Singh, S., Caicedo, J. & Carpenter, A. High-Dimensional Gene Expression and Morphology Profiles of Cells across 28,000 Genetic and Chemical Perturbations. *bioRxiv* 2021.09.08.459417 (2021)

[40] Way, G. P. et al. Morphology and gene expression profiling provide complementary information for mapping cell state. *bioRxiv* 2021.10.21.465335 (2021)

6. Please provide more details on the results of hyperparameter tuning and cross-validation in addition to those in Figures S12 and S13.

For the method related to hyperparameter tuning we now provide further details on Methods p29 :

“..., we used a grid search with balanced accuracy as the scoring function as implemented in scikit-learn, a Python based package, to optimise hyperparameters. Using a 4-fold stratified cross validation we determined the variation in performance when changing number of trees in the Random Forest. We used a GridSearchCV. The grid parameters varied the number of trees from 21 to 301 with a step size of 5. We checked for change in balanced accuracy in out of fold results in the cross validation and training time. No significant improvement was observed on increasing number of trees. Hence, we opted for the optimal baseline model with 100 trees which is within the optimal range of 64–128 number of trees given in previous research[77] has also shown that increasing the number of trees does not necessarily improve performance.[103]”

[77] Oshiro, T. M., Perez, P. S. & Baranauskas, J. A. How many trees in a random forest? *Lect. Notes Comput. Sci. (including Subser. Lect. Notes Artif. Intell. Lect. Notes Bioinformatics)* 7376 LNAI, 154–168 (2012)

[103] Jain, S., Kotsampasakou, E. & Ecker, G. F. Comparing the performance of meta-classifiers—a case study on selected imbalanced data sets relevant for prediction of liver toxicity. *J. Comput. Aided. Mol. Des.* 32, 583–590 (2018).

We have just double-checked the methods section regarding results from cross-validation for completeness and have assured that the text is complete as reported in Figure S7 (p.10 of supplementary information).

“When considering results from 200 test sets from the nested cross-validation, Morgan fingerprints (median F1 score of 0.42 and median precision of 0.35) perform similar to late-stage fusion combining Cell Painting, Gene Expression and Morgan Fingerprints (median F1 score of 0.42 and median precision of 0.32). However, compared to sensitivity of 0.53 when using Morgan fingerprints, the late-stage fusion combining Cell Painting, Gene Expression and Morgan Fingerprints recorded a median sensitivity of 0.63 (relatively increased by 17.2%, T-test p-value=1.5e-11). The corresponding median specificity decreased from 0.81 when Morgan fingerprints to 0.75 when using late-stage fusion combining Cell Painting, Gene Expression and Morgan Fingerprints (relatively decreased by 7.7%, T-test p-value=2.2e-05). When considering AUC-ROC, late-stage fusion combining Cell Painting, Gene Expression and Morgan Fingerprints (median AUC-ROC= 0.75) was significantly improved by 2.5% (T-test p-value=1.8e-09) compared to Morgan fingerprints (median AUC-ROC= 0.74). When considering mitotoxic classes only, AUC-PR for late-stage fusion combining Cell Painting, Gene Expression and Morgan Fingerprints (median AUC-PR = 0.49) was significantly improved by 6.7% (T-test p-value=6.8e-06) compared to Morgan fingerprints (median AUC-PR = 0.36).”

REVIEWERS' COMMENTS:

Reviewer #3 (Remarks to the Author):

The authors have addressed mostly my concerns.
One minor point is in figure 4, NA^+ should be Na^+ (superscript).